# Optimistic posterior sampling for reinforcement learning: worst-case regret bounds

**Shipra Agrawal**
Columbia University
sa3305@columbia.edu

**Randy Jia**
Columbia University
rqj2000@columbia.edu

## Abstract

We present an algorithm based on posterior sampling (aka Thompson sampling) that achieves near-optimal worst-case regret bounds when the underlying Markov Decision Process (MDP) is communicating with a finite, though unknown, diameter. Our main result is a high probability regret upper bound of $\tilde{O}(D\sqrt{SAT})$ for any communicating MDP with $S$ states, $A$ actions and diameter $D$, when $T \geq S^5 A$. Here, regret compares the total reward achieved by the algorithm to the total expected reward of an optimal infinite-horizon undiscounted average reward policy, in time horizon $T$. This result improves over the best previously known upper bound of $\tilde{O}(DS\sqrt{AT})$ achieved by any algorithm in this setting, and matches the dependence on $S$ in the established lower bound of $\Omega(\sqrt{DSAT})$ for this problem. Our techniques involve proving some novel results about the anti-concentration of Dirichlet distribution, which may be of independent interest.

## 1 Introduction

Reinforcement Learning (RL) refers to the problem of learning and planning in sequential decision making systems when the underlying system dynamics are unknown, and may need to be learned by trying out different options and observing their outcomes. A typical model for the sequential decision making problem is a Markov Decision Process (MDP), which proceeds in discrete time steps. At each time step, the system is in some state $s$, and the decision maker may take any available action $a$ to obtain a (possibly stochastic) reward. The system then transitions to the next state according to a fixed state transition distribution. The reward and the next state depend on the current state $s$ and the action $a$, but are independent of all the previous states and actions. In the reinforcement learning problem, the underlying state transition distributions and/or reward distributions are unknown, and need to be *learned* using the observed rewards and state transitions, while aiming to *maximize* the cumulative reward. This requires the algorithm to manage the tradeoff between exploration vs. exploitation, i.e., exploring different actions in different states in order to learn the model more accurately vs. taking actions that currently seem to be reward maximizing.

Exploration-exploitation tradeoff has been studied extensively in the context of stochastic multi-armed bandit (MAB) problems, which are essentially MDPs with a single state. The performance of MAB algorithms is typically measured through *regret*, which compares the total reward obtained by the algorithm to the total expected reward of an optimal action. Optimal regret bounds have been established for many variations of MAB (see Bubeck et al. [2012] for a survey), with a large majority of results obtained using the Upper Confidence Bound (UCB) algorithm, or more generally, the *optimism in the face of uncertainty* principle. Under this principle, the learning algorithm maintains tight over-estimates (or optimistic estimates) of the expected rewards for individual actions, and at any given step, picks the action with the highest optimistic estimate. More recently, posterior sampling, aka Thompson Sampling [Thompson, 1933], has emerged as another popular algorithm design principle in MAB, owing its popularity to a simple and extendible algorithmic structure, an

attractive empirical performance [Chapelle and Li, 2011, Kaufmann et al., 2012], as well as provably optimal performance bounds that have been recently obtained for many variations of MAB [Agrawal and Goyal, 2012, 2013b,a, Russo and Van Roy, 2015, 2014, Bubeck and Liu, 2013]. In this approach, the algorithm maintains a Bayesian posterior distribution for the expected reward of every action; then at any given step, it generates an independent sample from each of these posteriors and takes the action with the highest sample value.

We consider the reinforcement learning problem with finite states $S$ and finite actions $A$ in a similar regret based framework, where the total reward of the reinforcement learning algorithm is compared to the total expected reward achieved by a single benchmark policy over a time horizon $T$. In our setting, the benchmark policy is the *infinite-horizon undiscounted average reward* optimal policy for the underlying MDP, under the assumption that the MDP is communicating with (unknown) finite diameter $D$. The diameter $D$ is an upper bound on the time it takes to move from any state $s$ to any other state $s'$ using an appropriate policy, for each pair $s, s'$. A finite diameter is understood to be necessary for interesting bounds on the regret of any algorithm in this setting [Jaksch et al., 2010]. The UCRL2 algorithm of Jaksch et al. [2010], which is based on the optimism principle, achieved the best previously known upper bound of $\tilde{O}(DS\sqrt{AT})$ for this problem. A similar bound was achieved by Bartlett and Tewari [2009], though assuming the knowledge of the diameter $D$. Jaksch et al. [2010] also established a worst-case lower bound of $\Omega(\sqrt{DSAT})$ on the regret of any algorithm for this problem.

**Our main contribution** is a posterior sampling based algorithm with a high probability worst-case regret upper bound of $\tilde{O}(D\sqrt{SAT} + DS^{7/4}A^{3/4}T^{1/4})$, which is $\tilde{O}(D\sqrt{SAT})$ when $T \geq S^5 A$. This improves the previously best known upper bound for this problem by a factor of $\sqrt{S}$, and matches the dependence on $S$ in the lower bound, for large enough $T$.

Our algorithm uses an 'optimistic version' of the posterior sampling heuristic, while utilizing several ideas from the algorithm design structure in Jaksch et al. [2010], such as an epoch based execution and the extended MDP construction. The algorithm proceeds in epochs, where in the beginning of every epoch, it generates $\psi = \tilde{O}(S)$ sample transition probability vectors from a posterior distribution for every state and action, and solves an extended MDP with $\psi A$ actions and $S$ states formed using these samples. The optimal policy computed for this extended MDP is used throughout the epoch.

Posterior Sampling for Reinforcement Learning (PSRL) approach has been used previously in Osband et al. [2013], Abbasi-Yadkori and Szepesvari [2014], Osband and Van Roy [2016], but in a *Bayesian regret* framework. Bayesian regret is defined as the expected regret over a known prior on the transition probability matrix. Osband and Van Roy [2016] demonstrate an $\tilde{O}(H\sqrt{SAT})$ bound on the expected Bayesian regret for PSRL in finite-horizon episodic Markov decision processes, when the episode length is $H$. In this paper, we consider the stronger notion of *worst-case* regret, aka minimax regret, which requires bounding the maximum regret for any instance of the problem. [1] Further, we consider a non-episodic communicating MDP setting, and produce a comparable bound of $\tilde{O}(D\sqrt{SAT})$ for large $T$, where $D$ is the unknown diameter of the communicating MDP. In comparison to a single sample from the posterior in PSRL, our algorithm is slightly inefficient as it uses multiple ($\tilde{O}(S)$) samples. It is not entirely clear if the extra samples are only an artifact of the analysis. In an empirical study of a multiple sample version of posterior sampling for RL, Fonteneau et al. [2013] show that multiple samples can potentially improve the performance of posterior sampling in terms of probability of taking the optimal decision. Our analysis utilizes some ideas from the Bayesian regret analysis, most importantly the technique of stochastic optimism from Osband et al. [2014] for deriving tighter deviation bounds. However, bounding the worst-case regret requires several new technical ideas, in particular, for proving 'optimism' of the gain of the sampled MDP. Further discussion is provided in Section 4.

We should also compare our result with the very recent result of Azar et al. [2017], which provides an optimistic version of value-iteration algorithm with a minimax (i.e., worst-case) regret bound of

$\tilde{O}(\sqrt{HSAT})$ when $T \geq H^3 S^3 A$. However, the setting considered in Azar et al. [2017] is that of an *episodic MDP*, where the learning agent interacts with the system in episodes of fixed and known length $H$. The initial state of each episode can be arbitrary, but importantly, the sequence of these initial states is shared by the algorithm and any benchmark policy. In contrast, in the non-episodic setting considered in this paper, the state trajectory of the benchmark policy over $T$ time steps can be completely different from the algorithm's trajectory. To the best of our understanding, the shared sequence of initial states and the fixed known length $H$ of episodes seem to form crucial components of the analysis in Azar et al. [2017], making it difficult to extend their analysis to the non-episodic communicating MDP setting considered in this paper.

Among **other related work**, Burnetas and Katehakis [1997] and Tewari and Bartlett [2008] present optimistic linear programming approaches that achieve logarithmic regret bounds with problem dependent constants. Strong PAC bounds have been provided in Kearns and Singh [1999], Brafman and Tennenholtz [2002], Kakade et al. [2003], Asmuth et al. [2009], Dann and Brunskill [2015]. There, the aim is to bound the performance of the policy learned at the end of the learning horizon, and not the performance during learning as quantified here by regret. Notably, the BOSS algorithm proposed in Asmuth et al. [2009] is similar to the algorithm proposed here in the sense that the former also takes multiple samples from the posterior to form an extended (referred to as *merged*) MDP. Strehl and Littman [2005, 2008] provide an optimistic algorithm for bounding regret in a discounted reward setting, but the definition of regret is slightly different in that it measures the difference between the rewards of an optimal policy and the rewards of the learning algorithm along the *trajectory taken by the learning algorithm*.

## 2 Preliminaries and Problem Definition

### 2.1 Markov Decision Process (MDP)

We consider a Markov Decision Process $\mathcal{M}$ defined by tuple $\{\mathcal{S}, \mathcal{A}, P, r, s_1\}$, where $\mathcal{S}$ is a finite state-space of size $S$, $\mathcal{A}$ is a finite action-space of size $A$, $P : \mathcal{S} \times \mathcal{A} \to \Delta^{\mathcal{S}}$ is the transition model, $r : \mathcal{S} \times \mathcal{A} \to [0, 1]$ is the reward function, and $s_1$ is the starting state. When an action $a \in \mathcal{A}$ is taken in a state $s \in \mathcal{S}$, a reward $r_{s,a}$ is generated and the system transitions to the next state $s' \in \mathcal{S}$ with probability $P_{s,a}(s')$, where $\sum_{s' \in \mathcal{S}} P_{s,a}(s') = 1$.

We consider 'communicating' MDPs with finite 'diameter' (see Bartlett and Tewari [2009] for an in-depth discussion). Below we define communicating MDPs, and recall some useful known results for such MDPs.

**Definition 1** (Policy). *A deterministic policy $\pi : \mathcal{S} \to \mathcal{A}$ is a mapping from state space to action space.*

**Definition 2** (Diameter $D(\mathcal{M})$). *Diameter $D(\mathcal{M})$ of an MDP $\mathcal{M}$ is defined as the minimum time required to go from one state to another in the MDP using some deterministic policy:*

$$D(\mathcal{M}) = \max_{s \neq s', s, s' \in \mathcal{S}} \min_{\pi : \mathcal{S} \to \mathcal{A}} T^{\pi}_{s \to s'},$$

*where $T^{\pi}_{s \to s'}$ is the expected number of steps it takes to reach state $s'$ when starting from state $s$ and using policy $\pi$.*

**Definition 3** (Communicating MDP). *An MDP $\mathcal{M}$ is communicating if and only if it has a finite diameter. That is, for any two states $s \neq s'$, there exists a policy $\pi$ such that the expected number of steps to reach $s'$ from $s$, $T^{\pi}_{s \to s'}$, is at most $D$, for some finite $D \geq 0$.*

**Definition 4** (Gain of a policy). *The gain of a policy $\pi$, from starting state $s_1 = s$, is defined as the infinite horizon undiscounted average reward, given by*

$$\lambda^{\pi}(s) = \mathbb{E}[\lim_{T \to \infty} \frac{1}{T} \sum_{i=1}^{T} r_{s_t, \pi(s_t)} | s_1 = s].$$

*where $s_t$ is the state reached at time $t$.*

**Lemma 2.1** (Optimal gain for communicating MDPs). *For a communicating MDP $\mathcal{M}$ with diameter $D$:*

*(a) (Puterman [2014] Theorem 8.1.2, Theorem 8.3.2) The optimal (maximum) gain $\lambda^*$ is state independent and is achieved by a deterministic stationary policy $\pi^*$, i.e., there exists a deterministic policy $\pi^*$ such that*

$$\lambda^* := \max_{s' \in \mathcal{S}} \max_{\pi} \lambda^\pi(s') = \lambda^{\pi^*}(s), \forall s \in \mathcal{S}.$$

*Here, $\pi^*$ is referred to as an optimal policy for MDP $\mathcal{M}$.*

*(b) (Tewari and Bartlett [2008], Theorem 4) The optimal gain $\lambda^*$ satisfies the following equations,*

$$\lambda^* = \min_{h \in \mathbb{R}^S} \max_{s,a} r_{s,a} + P_{s,a}^T h - h_s = \max_a r_{s,a} + P_{s,a}^T h^* - h_s^*, \forall s \qquad (1)$$

*where $h^*$, referred to as the bias vector of MDP $\mathcal{M}$, satisfies:*

$$\max_s h_s^* - \min_s h_s^* \leq D.$$

Given the above definitions and results, we can now define the reinforcement learning problem studied in this paper.

## 2.2 The reinforcement learning problem

The reinforcement learning problem proceeds in rounds $t = 1, \dots, T$. The learning agent starts from a state $s_1$ at round $t = 1$. In the beginning of every round $t$, the agent takes an action $a_t \in \mathcal{A}$ and observes the reward $r_{s_t,a_t}$ as well as the next state $s_{t+1} \sim P_{s_t,a_t}$, where $r$ and $P$ are the reward function and the transition model, respectively, for a communicating MDP $\mathcal{M}$ with diameter $D$.

The learning agent knows the state-space $\mathcal{S}$, the action space $\mathcal{A}$, as well as the rewards $r_{s,a}, \forall s \in \mathcal{S}, a \in \mathcal{A}$, for the underlying MDP, but not the transition model $P$ or the diameter $D$. (The assumption of known and deterministic rewards has been made here only for simplicity of exposition, since the unknown transition model is the main source of difficulty in this problem. Our algorithm and results can be extended to bounded stochastic rewards with unknown distributions using standard Thompson Sampling for MAB, e.g., using the techniques in Agrawal and Goyal [2013b].)

The agent can use the past observations to learn the underlying MDP model and decide future actions. The goal is to maximize the total reward $\sum_{t=1}^{T} r_{s_t,a_t}$, or equivalently, minimize the total regret over a time horizon $T$, defined as

$$\mathcal{R}(T, \mathcal{M}) := T\lambda^* - \sum_{t=1}^{T} r_{s_t,a_t} \qquad (2)$$

where $\lambda^*$ is the optimal gain of MDP $\mathcal{M}$.

We present an algorithm for the learning agent with a near-optimal upper bound on the regret $\mathcal{R}(T, \mathcal{M})$ for any communicating MDP $\mathcal{M}$ with diameter $D$, thus bounding the worst-case regret over this class of MDPs.

# 3 Algorithm Description

Our algorithm combines the ideas of Posterior sampling (aka Thompson Sampling) with the extended MDP construction used in Jaksch et al. [2010]. Below we describe the main components of our algorithm.

*Some notations:* $N_{s,a}^t$ denotes the total number of times the algorithm visited state $s$ and played action $a$ until before time $t$, and $N_{s,a}^t(i)$ denotes the number of time steps among these $N_{s,a}^t$ steps where the next state was $i$, i.e., a transition from state $s$ to $i$ was observed. We index the states from 1 to $S$, so that $\sum_{i=1}^{S} N_{s,a}^t(i) = N_{s,a}^t$ for any $t$. We use the symbol $\mathbf{1}$ to denote the vector of all 1s, and $\mathbf{1}_i$ to denote the vector with 1 at the $i^{th}$ coordinate and 0 elsewhere.

**Doubling epochs:** Our algorithm uses the epoch based execution framework of Jaksch et al. [2010]. An epoch is a group of consecutive rounds. The rounds $t = 1, \dots, T$ are broken into consecutive epochs as follows: the $k^{th}$ epoch begins at the round $\tau_k$ immediately after the end of $(k-1)^{th}$ epoch and ends at the first round $\tau$ such that for some state-action pair $s, a$, $N_{s,a}^\tau \geq 2N_{s,a}^{\tau_k}$. The algorithm computes a new policy $\tilde{\pi}_k$ at the beginning of every epoch $k$, and uses that policy through all the rounds in that epoch. It is easy to observe that irrespective of how the policy $\tilde{\pi}_k$ is computed, the number of epochs in $T$ rounds is bounded by $SA \log(T)$.

**Posterior Sampling:** We use posterior sampling to compute the policy $\tilde{\pi}_k$ in the beginning of every epoch. Dirichlet distribution is a convenient choice maintaining posteriors for the transition probability vectors $P_{s,a}$ for every $s \in \mathcal{S}, a \in \mathcal{A}$, as they satisfy the following useful property: given a prior Dirichlet$(\alpha_1, \ldots, \alpha_S)$ on $P_{s,a}$, after observing a transition from state $s$ to $i$ (with underlying probability $P_{s,a}(i)$), the posterior distribution is given by Dirichlet$(\alpha_1, \ldots, \alpha_i + 1, \ldots, \alpha_S)$. By this property, for any $s \in \mathcal{S}, a \in \mathcal{A}$, on starting from prior Dirichlet($\mathbf{1}$) for $P_{s,a}$, the posterior at time $t$ is Dirichlet$(\{N_{s,a}^t(i) + 1\}_{i=1,\ldots,S})$.

Our algorithm uses a modified, optimistic version of this approach. At the beginning of every epoch $k$, for every $s \in \mathcal{S}, a \in \mathcal{A}$ such that $N_{s,a} \geq \eta$, it generates multiple samples for $P_{s,a}$ from a 'boosted' posterior. Specifically, it generates $\psi = O(S \log(SA/\rho))$ independent sample probability vectors $Q_{s,a}^{1,k}, \ldots, Q_{s,a}^{\psi,k}$ as

$$Q_{s,a}^{j,k} \sim \text{Dirichlet}(\mathbf{M}_{s,a}^{\tau_k}),$$

where $\mathbf{M}_{s,a}^t$ denotes the vector $[M_{s,a}^t(i)]_{i=1,\ldots,S}$, with

$$M_{s,a}^t(i) := \tfrac{1}{\kappa}(N_{s,a}^t(i) + \omega), \text{ for } i = 1, \ldots, S. \tag{3}$$

Here, $\kappa = O(\log(T/\rho))$, $\omega = O(\log(T/\rho))$, $\eta = \sqrt{\frac{TS}{A}} + 12\omega S^2$, and $\rho \in (0,1)$ is a parameter of the algorithm. In the regret analysis, we derive sufficiently large constants that can be used in the definition of $\psi, \kappa, \omega$ to guarantee the bounds. However, no attempt has been made to optimize those constants, and it is likely that much smaller constants suffice.

For every remaining $s, a$, i.e., those with small $N_{s,a}$ ($N_{s,a} < \eta$) the algorithm use a simple optimistic sampling described in Algorithm 1. This special sampling for $s, a$ with small $N_{s,a}$ has been introduced to handle a technical difficulty in analyzing the anti-concentration of Dirichlet posteriors when the parameters are very small. We suspect that with an improved analysis, this may not be required.

**Extended MDP:** The policy $\tilde{\pi}_k$ to be used in epoch $k$ is computed as the optimal policy of an *extended MDP* $\tilde{\mathcal{M}}^k$ defined by the sampled transition probability vectors, using the construction of Jaksch et al. [2010]. Given sampled vectors $Q_{s,a}^{j,k}, j = 1, \ldots, \psi$, for every state-action pair $s, a$, we define extended MDP $\tilde{\mathcal{M}}^k$ by extending the original action space as follows: for every $s, a$, create $\psi$ actions for every action $a \in A$, denoting by $a^j$ the action corresponding to action $a$ and sample $j$; then, in MDP $\tilde{\mathcal{M}}^k$, on taking action $a^j$ in state $s$, reward is $r_{s,a}$ but transitions to the next state follows the transition probability vector $Q_{s,a}^{j,k}$.

Note that the algorithm uses the optimal policy $\tilde{\pi}_k$ of extended MDP $\tilde{\mathcal{M}}^k$ to take actions in the action space $\mathcal{A}$ which is technically different from the action space of MDP $\tilde{\mathcal{M}}^k$, where the policy $\tilde{\pi}_k$ is defined. We slightly abuse the notation to say that the algorithm takes action $a_t = \tilde{\pi}(s_t)$ to mean that the algorithm takes action $a_t = a \in \mathcal{A}$ when $\tilde{\pi}_k(s_t) = a^j$ for some $j$.

Our algorithm is summarized as Algorithm 1.

## 4 Regret Bounds

We prove the following bound on the regret of Algorithm 1 for the reinforcement learning problem.

**Theorem 1.** *For any communicating MDP $\mathcal{M}$ with $S$ states, $A$ actions, and diameter $D$, with probability $1 - \rho$. the regret of Algorithm 1 in time $T \geq CDA \log^2(T/\rho)$ is bounded as:*

$$\mathcal{R}(T, \mathcal{M}) \leq \tilde{O}\left(D\sqrt{SAT} + DS^{7/4}A^{3/4}T^{1/4} + DS^{5/2}A\right)$$

*where $C$ is an absolute constant. For $T \geq S^5 A$, this implies a regret bound of*

$$\mathcal{R}(T, \mathcal{M}) \leq \tilde{O}\left(D\sqrt{SAT}\right).$$

*Here $\tilde{O}$ hides logarithmic factors in $S, A, T, \rho$ and absolute constants.*

The rest of this section is devoted to proving the above theorem. Here, we provide a sketch of the proof and discuss some of the key lemmas, all missing details are provided in the supplementary material.

---

**Algorithm 1** A posterior sampling based algorithm for the reinforcement learning problem

---

**Inputs:** State space $\mathcal{S}$, Action space $\mathcal{A}$, starting state $s_1$, reward function $r$, time horizon $T$, parameters $\rho \in (0,1], \psi = O(S\log(SA/\rho)), \omega = O(\log(T/\rho)), \kappa = O(\log(T/\rho)), \eta = \sqrt{\frac{TS}{A} + 12\omega S^2}$.

**Initialize:** $\tau^1 := 1, \mathbf{M}_{s,a}^{\tau_1} = \omega \mathbf{1}$.

**for all** epochs $k = 1, 2, \ldots,$ **do**

> *Sample transition probability vectors:* For each $s, a$, generate $\psi$ independent sample probability vectors $Q_{s,a}^{j,k}, j = 1, \ldots, \psi$, as follows:
>
> - **(Posterior sampling):** For $s, a$ such that $N_{s,a}^{\tau_k} \geq \eta$, use samples from the Dirichlet distribution:
> $$Q_{s,a}^{j,k} \sim \text{Dirichlet}(\mathbf{M}_{s,a}^{\tau_k}),$$
>
> - **(Simple optimistic sampling):** For remaining $s, a$, with $N_{s,a}^{\tau_k} < \eta$, use the following simple optimistic sampling: let
> $$P_{s,a}^- = \hat{P}_{s,a} - \mathbf{\Delta},$$
> where $\hat{P}_{s,a}(i) = \frac{N_{s,a}^{\tau_k}(i)}{N_{s,a}^{\tau_k}}$, and $\Delta_i = \min\left\{\sqrt{\frac{3\hat{P}_{s,a}(i)\log(4S)}{N_{s,a}^{\tau_k}}} + \frac{3\log(4S)}{N_{s,a}^{\tau_k}}, \hat{P}_{s,a}(i)\right\}$, and let $\mathbf{z}$ be a random vector picked uniformly at random from $\{\mathbf{1}_1, \ldots, \mathbf{1}_S\}$; set
> $$Q_{s,a}^{j,k} = P_{s,a}^- + (1 - \textstyle\sum_{i=1}^{S} P_{s,a}^-(i))\mathbf{z}.$$
>
> *Compute policy $\tilde{\pi}^k$:* as the optimal gain policy for extended MDP $\tilde{\mathcal{M}}^k$ constructed using sample set $\{Q_{s,a}^{j,k}, j = 1, \ldots, \psi, s \in \mathcal{S}, a \in \mathcal{A}\}$.
>
> *Execute policy $\tilde{\pi}^k$:*
> **for all** time steps $t = \tau_k, \tau_k + 1, \ldots,$ until `break epoch` **do**
> > Play action $a_t = \tilde{\pi}_k(s_t)$.
> > Observe the transition to the next state $s_{t+1}$.
> > Set $N_{s,a}^{t+1}(i), M_{s,a}^{t+1}(i)$ for all $a \in \mathcal{A}, s, i \in \mathcal{S}$ as defined (refer to Equation (3)).
> > If $N_{s_t,a_t}^{t+1} \geq 2N_{s_t,a_t}^{\tau_k}$, then set $\tau_{k+1} = t + 1$ and `break epoch`.
> **end for**
> **end for**

---

### 4.1 Proof of Theorem 1

As defined in Section 2, regret $\mathcal{R}(T, \mathcal{M})$ is given by $\mathcal{R}(T, \mathcal{M}) = T\lambda^* - \sum_{t=1}^{T} r_{s_t, a_t}$, where $\lambda^*$ is the optimal gain of MDP $\mathcal{M}$, $a_t$ is the action taken and $s_t$ is the state reached by the algorithm at time $t$. Algorithm 1 proceeds in epochs $k = 1, 2, \ldots, K$, where $K \leq SA\log(T)$. To bound its regret in time $T$, we first analyze the regret in each epoch $k$, namely,

$$\mathcal{R}_k := (\tau_{k+1} - \tau_k)\lambda^* - \sum_{t=\tau_k}^{\tau_{k+1}-1} r_{s_t, a_t},$$

and bound $\mathcal{R}_k$ by roughly

$$D \sum_{s,a} \frac{N_{s,a}^{\tau_{k+1}} - N_{s,a}^{\tau_k}}{\sqrt{N_{s,a}^{\tau_k}}}$$

where, by definition, for every $s, a$, $(N_{s,a}^{\tau_{k+1}} - N_{s,a}^{\tau_k})$ is the number of times this state-action pair is visited in epoch $k$. The proof of this bound has two main components:

(a) **Optimism:** The policy $\tilde{\pi}_k$ used by the algorithm in epoch $k$ is computed as an optimal gain policy of the extended MDP $\tilde{\mathcal{M}}^k$. The first part of the proof is to show that with high probability, the extended MDP $\tilde{\mathcal{M}}^k$ is (i) a communicating MDP with diameter at most $2D$, and (ii) optimistic, i.e., has optimal gain at least (close to) $\lambda^*$. Part (i) is stated as Lemma 4.1, with a proof provided in the supplementary material. Now, let $\tilde{\lambda}_k$ be the optimal gain of the extended MDP $\tilde{\mathcal{M}}^k$. In

Lemma 4.2, which forms one of the main novel technical components of our proof, we show that with probability $1 - \rho$,

$$\tilde{\lambda}_k \geq \lambda^* - \tilde{O}(D\sqrt{\tfrac{SA}{T}}).$$

We first show that above holds if for every $s, a$, there exists a sample transition probability vector whose projection on a *fixed* unknown vector ($h^*$) is optimistic. Then, in Lemma 4.3 we prove this optimism by deriving a fundamental new result on the anti-concentration of any fixed projection of a Dirichlet random vector (Proposition A.1 in the supplementary material).

Substituting this upper bound on $\lambda^*$, we have the following bound on $\mathcal{R}_k$ with probability $1 - \rho$:

$$\mathcal{R}_k \leq \sum_{t=\tau_k}^{\tau_{k+1}-1} \left( \tilde{\lambda}_k - r_{s_t, a_t} + \tilde{O}(D\sqrt{\tfrac{SA}{T}}) \right). \tag{4}$$

(b) **Deviation bounds:** Optimism guarantees that with high probability, the optimal gain $\tilde{\lambda}_k$ for MDP $\tilde{\mathcal{M}}^k$ is at least $\lambda^*$. And, by definition of $\tilde{\pi}_k$, $\tilde{\lambda}_k$ is the gain of the chosen policy $\tilde{\pi}_k$ for MDP $\tilde{\mathcal{M}}^k$. However, the algorithm executes this policy on the true MDP $\mathcal{M}$. The only difference between the two is the transition model: on taking an action $a^j := \tilde{\pi}_k(s)$ in state $s$ in MDP $\tilde{\mathcal{M}}^k$, the next state follows the sampled distribution

$$\tilde{P}_{s,a} := Q_{s,a}^{j,k}, \tag{5}$$

where as on taking the corresponding action $a$ in MDP $\mathcal{M}$, the next state follows the distribution $P_{s,a}$. The next step is to bound the difference between $\tilde{\lambda}_k$ and the average reward obtained by the algorithm by bounding the deviation $(\tilde{P}_{s,a} - P_{s,a})$. This line of argument bears similarities to the analysis of UCRL2 in Jaksch et al. [2010], with tighter deviation bounds that we are able to guarantee due to the use of posterior sampling instead of deterministic optimistic bias used in UCRL2. Now, since $a_t = \tilde{\pi}_k(s_t)$, using the relation between the gain $\tilde{\lambda}_k$, the bias vector $\tilde{h}$, and reward vector of optimal policy $\tilde{\pi}_k$ for communicating MDP $\tilde{\mathcal{M}}^k$ (refer to Lemma 2.1)

$$\sum_{t=\tau_k}^{\tau_{k+1}-1} \left( \tilde{\lambda} - r_{s_t, a_t} \right) = \sum_{t=\tau_k}^{\tau_{k+1}-1} (\tilde{P}_{s_t, a_t} - \mathbf{1}_{s_t})^T \tilde{h}$$

$$= \sum_{t=\tau_k}^{\tau_{k+1}-1} (\tilde{P}_{s_t, a_t} - P_{s_t, a_t} + P_{s_t, a_t} - \mathbf{1}_{s_t})^T \tilde{h} \tag{6}$$

where with high probability, $\tilde{h} \in \mathbb{R}^S$, the bias vector of MDP $\tilde{\mathcal{M}}^k$ satisfies

$$\max_s \tilde{h}_s - \min_s \tilde{h}_s \leq D(\tilde{\mathcal{M}}^k) \leq 2D \text{ (refer to Lemma 4.1).}$$

Next, we bound the deviation $(\tilde{P}_{s,a} - P_{s,a})^T \tilde{h}$ for all $s, a$, to bound the first term in above. Note that $\tilde{h}$ is random and can be arbitrarily correlated with $\tilde{P}$, therefore, we need to bound $\max_{h \in [0, 2D]^S} (\tilde{P}_{s,a} - P_{s,a})^T h$. (For the above term, w.l.o.g. we can assume $\tilde{h} \in [0, 2D]^S$).

For $s, a$ such that $N_{s,a}^{\tau_k} > \eta$, $\tilde{P}_{s,a} = Q_{s,a}^{j,k}$ is a sample from the Dirichlet posterior. In Lemma 4.4, we show that with high probability,

$$\max_{h \in [0, 2D]^S} (\tilde{P}_{s,a}^k - P_{s,a})^T h \leq \tilde{O}(\frac{D}{\sqrt{N_{s,a}^{\tau_k}}} + \frac{DS}{N_{s,a}^{\tau_k}}). \tag{7}$$

This bound is an improvement by a $\sqrt{S}$ factor over the corresponding deviation bound obtainable for the optimistic estimates of $\tilde{P}_{s,a}$ in UCRL2. The derivation of this bound utilizes and extends the stochastic optimism technique from Osband et al. [2014]. For $s, a$ with $N_{s,a}^{\tau_k} \leq \eta$, $\tilde{P}_{s,a} = Q_{s,a}^{j,k}$ is a sample from the simple optimistic sampling, where we can only show the following weaker bound, but since this is used only while $N_{s,a}^{\tau_k}$ is small, the total contribution of this deviation will be small:

$$\max_{h \in [0, 2D]^S} (\tilde{P}_{s,a}^k - P_{s,a})^T h \leq \tilde{O}\left( D\sqrt{\frac{S}{N_{s,a}^{\tau_k}}} + \frac{DS}{N_{s,a}^{\tau_k}} \right). \tag{8}$$

Finally, to bound the second term in (6), we observe that $\mathbb{E}[\mathbf{1}_{s_{t+1}}^T \tilde{h} | \tilde{\pi}_k, \tilde{h}, s_t] = P_{s_t, a_t}^T \tilde{h}$ and use Azuma-Hoeffding inequality to obtain with probability $(1 - \frac{\rho}{SA})$:

$$\sum_{t=\tau_k}^{\tau_{k+1}-1} (P_{s_t, a_t} - \mathbf{1}_{s_t})^T \tilde{h} \leq O(\sqrt{(\tau_{k+1} - \tau_k) \log(SA/\rho)}). \tag{9}$$

Combining the above observations (equations (4), (6), (7), (8), (9)), we obtain the following bound on $\mathcal{R}_k$ within logarithmic factors:

$$D(\tau_{k+1}-\tau_k)\sqrt{\frac{SA}{T}}+D\sum_{s,a}\frac{N_{s,a}^{\tau_{k+1}}-N_{s,a}^{\tau_k}}{\sqrt{N_{s,a}^{\tau_k}}}\left(\mathbb{1}(N_{s,a}^{\tau_{k+1}}>\eta)+\sqrt{S}\mathbb{1}(N_{s,a}^{\tau_{k+1}}\leq\eta)\right)+D\sqrt{\tau_{k+1}-\tau_k}.$$

(10)

We can finish the proof by observing that (by definition of an epoch) the number of visits of any state-action pair can at most double in an epoch,

$$N_{s,a}^{\tau_{k+1}}-N_{s,a}^{\tau_k}\leq N_{s,a}^{\tau_k},$$

and therefore, substituting this observation in (10), we can bound (within logarithmic factors) the total regret $\mathcal{R}(T)=\sum_{k=1}^K\mathcal{R}_k$ as:

$$\sum_{k=1}^K\left(D(\tau_{k+1}-\tau_k)\sqrt{\frac{SA}{T}}+D\sum_{s,a:N_{s,a}^{\tau_k}>\eta}\sqrt{N_{s,a}^{\tau_k}}+D\sum_{s,a:N_{s,a}^{\tau_k}<\eta}\sqrt{SN_{s,a}^{\tau_k}}+D\sqrt{\tau_{k+1}-\tau_k}\right)$$
$$\leq\quad D\sqrt{SAT}+D\log(K)(\sum_{s,a}\sqrt{N_{s,a}^{\tau_K}})+D\log(K)(SA\sqrt{S\eta})+D\sqrt{KT}$$

where we used $N_{s,a}^{\tau_{k+1}}\leq 2N_{s,a}^{\tau_k}$ and $\sum_k(\tau_{k+1}-\tau_k)=T$. Now, we use that $K\leq SA\log(T)$, and $SA\sqrt{S\eta}=O(S^{7/4}A^{3/4}T^{1/4}+S^{5/2}A\log(T/\rho))$ (using $\eta=\sqrt{\frac{TS}{A}}+12\omega S^2$). Also, since $\sum_{s,a}N_{s,a}^{\tau_K}\leq T$, by simple worst scenario analysis, $\sum_{s,a}\sqrt{N_{s,a}^{\tau_K}}\leq\sqrt{SAT}$, and we obtain,

$$\mathcal{R}(T,\mathcal{M})\leq\tilde{O}(D\sqrt{SAT}+DS^{7/4}A^{3/4}T^{1/4}+DS^{5/2}A).$$

## 4.2 Main lemmas

Following lemma form the main technical components of our proof. All the missing proofs are provided in the supplementary material.

**Lemma 4.1.** *Assume $T\geq CDA\log^2(T/\rho)$ for a large enough constant $C$. Then, with probability $1-\rho$, for every epoch $k$, the diameter of MDP $\tilde{\mathcal{M}}^k$ is bounded by $2D$.*

**Lemma 4.2.** *With probability $1-\rho$, for every epoch $k$, the optimal gain $\tilde{\lambda}_k$ of the extended MDP $\tilde{\mathcal{M}}^k$ satisfies:*

$$\tilde{\lambda}_k\geq\lambda^*-O\left(D\log^2(T/\rho)\sqrt{\frac{SA}{T}}\right),$$

*where $\lambda^*$ the optimal gain of MDP $\mathcal{M}$ and $D$ is the diameter.*

*Proof.* Let $h^*$ be the bias vector for an optimal policy $\pi^*$ of MDP $\mathcal{M}$ (refer to Lemma 2.1 in the preliminaries section). Since $h^*$ is a fixed (though unknown) vector with $|h_i-h_j|\leq D$, we can apply Lemma 4.3 to obtain that with probability $1-\rho$, for all $s,a$, there exists a sample vector $Q_{s,a}^{j,k}$ for some $j\in\{1,\ldots,\psi\}$ such that

$$(Q_{s,a}^{j,k})^T h^*\geq P_{s,a}^T h^*-\delta$$

where $\delta=O\left(D\log^2(T/\rho)\sqrt{\frac{SA}{T}}\right)$. Now, consider the policy $\pi$ for MDP $\tilde{\mathcal{M}}^k$ which for any $s$, takes action $a^j$, with $a=\pi^*(s)$ and $j$ being a sample satisfying above inequality. Let $Q_\pi$ be the transition matrix for this policy, whose rows are formed by the vectors $Q_{s,\pi^*(s)}^{j,k}$, and $P_{\pi^*}$ be the transition matrix whose rows are formed by the vectors $P_{s,\pi^*(s)}$. Above implies $Q_\pi h^*\geq P_{\pi^*}h^*-\delta\mathbf{1}$. We use this inequality along with the known relations between the gain and the bias of optimal policy in communicating MDPs to obtain that the gain $\tilde{\lambda}(\pi)$ of policy in $\pi$ for MDP $\tilde{\mathcal{M}}^k$ satisfies $\tilde{\lambda}(\pi)\geq\lambda^*-\delta$ (details provided in the supplementary material), which proves the lemma statement since by optimality $\tilde{\lambda}_k\geq\tilde{\lambda}(\pi)$. $\square$

**Lemma 4.3.** (**Optimistic Sampling**) *Fix any vector $h \in \mathbb{R}^S$ such that $|h_i - h_{i'}| \leq D$ for any $i, i'$, and any epoch $k$. Then, for every $s, a$, with probability $1 - \frac{\rho}{SA}$ there exists at least one $j$ such that*

$$(Q_{s,a}^{j,k})^T h \geq P_{s,a}^T h - O\left(D \log^2(T/\rho)\sqrt{\frac{SA}{T}}\right).$$

**Lemma 4.4.** (**Deviation bound**) *With probability $1 - \rho$, for all epochs $k$, sample $j$, all $s, a$*

$$\max_{h \in [0, 2D]^S} (Q_{s,a}^{j,k} - P_{s,a})^T h \leq \begin{cases} O\left(D\sqrt{\dfrac{\log(SAT/\rho)}{N_{s,a}^{\tau_k}}} + D\dfrac{S\log(SAT/\rho)}{N_{s,a}^{\tau_k}}\right), & N_{s,a}^{\tau_k} > \eta \\ O\left(D\sqrt{\dfrac{S\log(SAT/\rho)}{N_{s,a}^{\tau_k}}} + D\dfrac{S\log(S)}{N_{s,a}^{\tau_k}}\right), & N_{s,a}^{\tau_k} \leq \eta \end{cases}$$

## 5 Conclusions

We presented an algorithm inspired by posterior sampling that achieves near-optimal worst-case regret bounds for the reinforcement learning problem with communicating MDPs in a non-episodic, undiscounted average reward setting. Our algorithm may be viewed as a more efficient randomized version of the UCRL2 algorithm of Jaksch et al. [2010], with randomization via posterior sampling forming the key to the $\sqrt{S}$ factor improvement in the regret bound provided by our algorithm. Our analysis demonstrates that posterior sampling provides the right amount of uncertainty in the samples, so that an optimistic policy can be obtained without excess over-estimation.

While our work surmounts some important technical difficulties in obtaining worst-case regret bounds for posterior sampling based algorithms for communicating MDPs, the provided bound is tight in its dependence on $S$ and $A$ only for large $T$ (specifically, for $T \geq S^5 A$). Other related results on tight worst-case regret bounds have a similar requirement of large $T$ (Azar et al. [2017] produce an $\tilde{O}(\sqrt{HSAT})$ bound when $T \geq H^3 S^3 A$). Obtaining a cleaner worst-case regret bound that does not require such a condition remains an open question. Other important directions of future work include reducing the number of posterior samples required in every epoch from $\tilde{O}(S)$ to constant or logarithmic in $S$, and extensions to contextual and continuous state MDPs.

## Footnotes

[1]Worst-case regret is a strictly stronger notion of regret in case the reward distribution function is known and only the transition probability distribution is unknown, as we will assume here for the most part. In case of unknown reward distribution, extending our worst-case regret bounds would require an assumption of bounded rewards, where as the Bayesian regret bounds in the above-mentioned literature allow more general (known) priors on the reward distributions with possibly unbounded support. Bayesian regret bounds in those more general settings are incomparable to the worst-case regret bounds presented here.

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
