[Supplementary Material · nips_paper787_supp.pdf]



**Organization.** In Section A, we prove some novel results about anti-concentration of Dirichlet random vectors. These are used in Section B to prove Lemma 4.2 and Lemma 4.3. In Section C, we prove several concentration bounds on Dirichlet posteriors and empirical estimates of transition probability vectors to prove Lemma 4.4. Here, we utilize the stochastic optimism technique from Osband et al. [2014]. In Section D, we prove Lemma 4.1 bounding the diameter of extended MDP with high probability. And, in Section E we list some known results (or easy corollaries of known results) that are utilized in our proofs.

## A   Anti-concentration of Dirichlet distribution

We prove the following general result on anti-concentration of Dirichlet distributions, which will be used to prove optimism.

**Proposition A.1.** *Consider a random vector $\tilde{p}$ generated from Dirichlet distribution with parameters $(m\bar{p}_1, \ldots, m\bar{p}_S)$, where $m\bar{p}_i \geq 6$. Then, for any* **fixed** *$h \in [0, D]^S$, with probability $\Omega(1/S) - S\rho$,*

$$(\tilde{p} - \bar{p})^T h \geq \frac{1}{8}\sqrt{\sum_{i<S} \frac{\bar{\gamma}_i \bar{c}_i^2}{m} - \frac{2SD\log(2/\rho)}{m}}$$

*where*

$$\bar{\gamma}_i := \frac{\bar{p}_i(\bar{p}_{i+1} + \ldots + \bar{p}_S)}{(\bar{p}_i + \ldots + \bar{p}_S)}, \bar{c}_i = (h_i - \bar{H}_{i+1}), \bar{H}_{i+1} = \frac{1}{\sum_{j=i+1}^S \bar{p}_j} \sum_{j=i+1}^S h_j \bar{p}_j.$$

We use an equivalent representation of a Dirichlet vector in terms of independent Beta random variables.

**Fact 1.** *Fix an ordering of indices $1, \ldots, S$, and define $\tilde{y}_i := \frac{\tilde{p}_i}{\tilde{p}_i + \cdots + \tilde{p}_S}, \bar{y}_i := \frac{\bar{p}_i}{\bar{p}_i + \cdots + \bar{p}_S}$. Then, for any $h \in \mathbb{R}^S$,*

$$(\tilde{p} - \bar{p})^T h = \sum_i (\tilde{y}_i - \bar{y}_i)(h_i - \tilde{H}_{i+1})(\bar{p}_i + \cdots + \bar{p}_S) = \sum_i (\tilde{y}_i - \bar{y}_i)(h_i - \bar{H}_{i+1})(\tilde{p}_i + \cdots + \tilde{p}_S)$$

*where $\tilde{H}_{i+1} = \frac{1}{\sum_{j=i+1}^S \tilde{p}_j} \sum_{j=i+1}^S h_j \tilde{p}_j$, $\bar{H}_{i+1} = \frac{1}{\sum_{j=i+1}^S \bar{p}_j} \sum_{j=i+1}^S h_j \bar{p}_j$.*

**Fact 2.** *For $i = 1, \ldots, S$, $\tilde{y}_i := \frac{\tilde{p}_i}{\tilde{p}_i + \cdots + \tilde{p}_S}$ are independent Beta random variables distributed as $Beta(m\bar{p}_i, m(\bar{p}_{i+1} + \cdots + \bar{p}_S))$, with mean*

$$\mathbb{E}[\tilde{y}_i] = \frac{m\bar{p}_i}{m(\bar{p}_i + \cdots + \bar{p}_S)} = \bar{y}_i,$$

*and variance*

$$\bar{\sigma}_i^2 := \mathbb{E}[(\tilde{y}_i - \bar{y}_i)^2] = \frac{\bar{p}_i(\bar{p}_{i+1} + \cdots + \bar{p}_S)}{(\bar{p}_i + \cdots + \bar{p}_S)^2(m(\bar{p}_i + \cdots + \bar{p}_S) + 1)}.$$

**Lemma A.2** (Corollary of Lemma E.2). *Let $\tilde{y}_i, \bar{y}_i, \bar{\sigma}_i$ be defined as in Fact 2. If $m\bar{p}_i, m(\bar{p}_{i+1} + \cdots + \bar{p}_S) \geq 6$, then, for any positive constant $C \leq \frac{1}{2}$,*

$$P(\tilde{y}_i \geq \bar{y}_i + C\bar{\sigma}_i + \frac{C}{m(\bar{p}_i + \ldots + \bar{p}_S)}) \geq 0.15 =: \eta.$$

*Proof.* Apply Lemma E.2 with $a = m\bar{p}_i, b = m(\bar{p}_{i+1} + \cdots + \bar{p}_S)$. $\qquad\square$

**Lemma A.3.** *(Application of Berry-Esseen theorem) Let $G \subseteq \{1, \ldots, S\}$ be a set of indices, $z_i \in \mathbb{R}$ be fixed. Let*

$$X_G := \sum_{i \in G} (\tilde{y}_i - \bar{y}_i)z_i.$$

*Let $F$ be the cumulative distribution function of*

$$\frac{X_G}{\sigma_G}, \text{ where, } \sigma_G^2 = \sum_{i \in G} z_i^2 \bar{\sigma}_i^2,$$

$\bar{\sigma}_i$ *being the standard deviation of $\tilde{y}_i$ (refer to Fact 2). Let $\Phi$ be the cumulative distribution function of standard normal distribution. Then, for all $\epsilon > 0$:*

$$\sup_x |F(x) - \Phi(x)| \le \epsilon$$

*as long as*

$$\sqrt{|G|} \ge \frac{RC}{\epsilon}, \text{ where } R := \max_{i,j \in G} \frac{z_i \bar{\sigma}_i}{z_j \bar{\sigma}_j}$$

*for some $C \le 3 + \frac{6}{m\bar{p}_i}$.*

*Proof.* $Y_i = (\tilde{y}_i - \bar{y}_i)z_i$. Then, $Y_i, i \in G$ are independent variables, with $\mathbb{E}[Y_i] = 0$,

$$\begin{aligned}
\sigma_i^2 := \mathbb{E}[Y_i^2] &= \mathbb{E}[(\tilde{y}_i - \bar{y}_i)^2 (z_i)^2] \\
&= z_i^2 \bar{\sigma}_i^2
\end{aligned}$$

$$\begin{aligned}
\rho_i := \mathbb{E}[|Y_i|^3] &\le \mathbb{E}[|Y_i|^4]^{3/4} \\
&= \mathbb{E}[|\tilde{y} - \bar{y}|^4]^{3/4} z_i^3 \\
&\le \kappa \mathbb{E}[|\tilde{y} - \bar{y}|^2]^{3/2} z_i^3 \\
&= \kappa \bar{\sigma}_i^3 z_i^3
\end{aligned}$$

where the first inequality is by using Jensen's inequality and $\kappa$ is the Kurtosis of Beta distribution. Next, we use that $\tilde{y}$ is Beta distributed, and Kurtosis of $Beta(\nu\mu, \nu(1-\mu))$ Distribution is

$$\kappa = 3 + \frac{6}{(3+\nu)} \left( \frac{(1-2\mu)^2(1+\nu)}{\mu(1-\mu)(2+\nu)} - 1 \right) \le 3 + \frac{6}{(3+\nu)\mu}.$$

Here, $\alpha = m(\bar{p}_i + \cdots + \bar{p}_S)\bar{y}_i, \beta = m(\bar{p}_i + \cdots + \bar{p}_S)(1 - \bar{y}_i)$, so that

$$\kappa \le 3 + \frac{6}{3 + m(\bar{p}_i + \cdots + \bar{p}_S)} \frac{1}{\bar{y}_i} \le 3 + \frac{6}{m\bar{p}_i}.$$

Now, we use Berry-Esseen theorem (Fact 6), with

$$\begin{aligned}
\psi_1 &= \frac{1}{\sqrt{\sum_{i \in G} \sigma_i^2}} \max_{i \in G} \frac{\rho_i}{\sigma_i^2} \\
&\le \frac{\kappa}{\sqrt{|G|}} \frac{\max_{i \in G} z_i \bar{\sigma}_i}{\min_{i \in G} z_i \bar{\sigma}_i}
\end{aligned}$$

to obtain the lemma statement.

$\square$

**Lemma A.4.** *Assuming $m\bar{p}_i \ge 6, \forall i$, for any fixed $z_i$, $i = 1, \ldots, S$,*

$$\Pr\left( \sum_i (\tilde{y}_i - \bar{y}_i)z_i \ge \frac{1}{4} \sqrt{\sum_i \bar{\sigma}_i^2 z_i^2} \right) \ge \Omega(1/S).$$

*Proof.* Define constant $\delta := \frac{(1-\Phi)(\frac{1}{2})}{2}$ and $k(\delta) := \frac{C^2}{\delta^4}$, where $C \le 4$.

Consider the the group of indices with the $k(\delta)$ largest values of $|z_i \bar{\sigma}_i|$, call it group $G(1)$, and then divide the remaining into smallest possible collection $\mathcal{G}$ of groups such that $|z_i \bar{\sigma}_i|/|z_j \bar{\sigma}_j| \le \frac{1}{\delta}$ for all $i, j$ in any given group $G$. Define an ordering $\prec$ on groups by ordering them by maximum value of

$|z_i\bar{\sigma}_i|$ in the group. That is $G \succ G'$ if $\max_{i \in G} z_i^2 \bar{\sigma}_i^2 \geq \max_{j \in G'} z_j^2 \bar{\sigma}_j^2$ Note that by construction, for $G \succ G'$, we have $\max_{i \in G} z_i^2 \bar{\sigma}_i^2 \geq \frac{1}{\delta^2} \max_{j \in G'} z_j^2 \bar{\sigma}_j^2$.

Recall from Lemma A.3, for every group $G \in \mathcal{G}$ of size $\sqrt{|G|} > \frac{C}{\delta \epsilon}$, we have that its cdf is within $\epsilon$ of normal distribution cdf, giving that $\Pr(X_G \geq \frac{1}{2}\sigma_G) \geq 2\delta - \epsilon$. Using this result for $\epsilon = \delta$, we get that for every group of size at least $k(\delta)$, we have

$$\Pr(X_G \geq \frac{1}{2}\sigma_G) \geq \delta. \tag{11}$$

We will look at three types of the groups we created above:

- Top big groups: those among the top $\log_{1/\delta}(S)$ groups that have cardinality at least $k(\delta)$

- Top small groups: those among the top $\log_{1/\delta}(S)$ groups that have cardinality smaller than $k(\delta)$

- Bottom groups: those not among the top $\log_{1/\delta}(S)$ groups

Here, top groups refers to the those ranked higher according to the ordering $\succ$.

For the first group type above, apply (11) to obtain,

for all big groups among top $\log_{1/\delta}(S)$, $X_G \geq \frac{1}{2}\sigma_G$

$$\text{with probability at least } \delta^{\log_{1/\delta}(S)} = \frac{1}{S}. \tag{12}$$

Next, we analyze the remaining indices (among top small groups and bottom groups). Consider the group $G(1)$ we set aside. Using Lemma A.2 $k(\delta)$ times, we have:

$$\Pr\left(\sum_{i \in G(1)} (\tilde{y}_i - \bar{y}_i)z_i \geq 0.5\sqrt{\sum_{i \in G(1)} z_i^2 \bar{\sigma}_i^2}\right) \geq \eta^{k(\delta)}$$

where $\eta \geq 0.15$.

Now, if it is the case where the top group is of small size, we apply the above anticoncentration of beta for each element in the group, so that for all indices $i$ in this group, $(\tilde{y}_i - \bar{y}_i)z_i \geq 0.5z_i\bar{\sigma}_i$, with probability $\eta^{k(\delta)}$. To conclude, so far, we have with probability at least $\frac{1}{S}\eta^{2k(\delta)}$

$$\sum_{i \in G(1), i \in \text{top big groups}} (\tilde{y}_i - \bar{y}_i)z_i \geq 0.5\sqrt{\sum_{i \in G(1), i \in \text{top big groups}} z_i^2 \bar{\sigma}_i^2}.$$

For every other small group $G$, the group's total variance is at most $k(\delta)\max_{i \in G} z_i^2 \bar{\sigma}_i^2 \leq k(\delta)\delta^{2j} z_{(1)}^2 \bar{\sigma}_{(1)}^2$, where $j$ is the rank of the group in ordering $\succ$ and $(1)$ is the index of the smallest variance in $G(1)$. So, the sum of the standard deviation for top $\log_{1/\delta}(S)$ small groups is at most

$$k(\delta)\sum_{G:\text{top small groups}} \max_{i \in G} z_i^2 \bar{\sigma}_i^2 \leq k(\delta)\sum_{j=1}^{\log_{1/\delta}(S)} \delta^{2j} z_{(1)}\bar{\sigma}_{(1)} \leq \frac{k(\delta)\delta^2}{1 - \delta^2} z_{(1)}^2 \bar{\sigma}_{(1)}^2$$

as it is a geometric series with $\delta$ multiplier. For the remaining bottom group, each element's variance is at most $\frac{1}{S^2} z_{(1)}^2 \bar{\sigma}_{(1)}^2$, therefore

$$\sum_{i:\text{top small groups, bottom groups}} z_i^2 \bar{\sigma}_i^2 \leq (\frac{k(\delta)\delta^2}{1 - \delta^2} + \frac{1}{S})z_{(1)}^2 \bar{\sigma}_{(1)}^2 \leq \frac{k(\delta)}{25} z_{(1)}^2 \bar{\sigma}_{(1)}^2 \leq \frac{1}{25}\sum_{i \in G(1)} z_i^2 \bar{\sigma}_i^2.$$

By Cantelli's Inequality (Fact 5), with probability at least $\frac{1}{2}$,

$$\sum_{i:\text{top small groups, bottom groups}} (\tilde{y}_i - \bar{y}_i)z_i \geq -\sqrt{\sum_{i\in\text{top small groups, bottom groups}} z_i^2\bar{\sigma}_i^2} \geq -\frac{1}{5}\sqrt{\sum_{i\in\text{G}(1)} z_i^2\bar{\sigma}_i^2}.$$

Hence combining our results above,

$$
\begin{aligned}
\sum_i (\tilde{y}_i - \bar{y}_i)z_i &\geq \frac{1}{2}\sqrt{\sum_{i\in G(1),\text{top big groups}} z_i^2\bar{\sigma}_i^2} - \frac{1}{5}\sqrt{\sum_{i\in G(1)} z_i^2\bar{\sigma}_i^2} \\
&\geq \frac{3}{10}\sqrt{\sum_{i\in G(1),\text{top big groups}} z_i^2\bar{\sigma}_i^2} + \frac{1}{25}\sqrt{\sum_{i\in G(1)} z_i^2\bar{\sigma}_i^2} - \frac{1}{25}\sqrt{\sum_{i\in G(1)} z_i^2\bar{\sigma}_i^2} \\
&\geq \frac{13}{50}\sqrt{\sum_{i\in G(1),\text{top big groups}} z_i^2\bar{\sigma}_i^2} + \frac{1}{25}\sqrt{\sum_{i\in G(1)} z_i^2\bar{\sigma}_i^2} \\
&\geq \frac{1}{4}\sqrt{\sum_i z_i^2\bar{\sigma}_i^2}
\end{aligned}
$$

with probability $\eta^{2k(\delta)}\frac{1}{2S} = \Omega(1/S)$.

$\square$

*Proof.* **(Proof of Proposition A.1)** Use Fact 1 to express $(\tilde{p} - \bar{p})^T h$ as:

$$(\tilde{p} - \bar{p})^T h = \sum_i (\tilde{y}_i - \bar{y}_i)(h_i - \tilde{H}_{i+1})(\bar{p}_i + \cdots + \bar{p}_S).$$

Using Lemma E.3 and Corollary E.7,

$$|\tilde{H}_i - \bar{H}_i| \leq D\sqrt{\frac{2\log(2/\rho)}{m(\bar{p}_i + \ldots + \bar{p}_S)}}$$

with probability $1 - \rho$ for any $i$.

and similarly using Lemma E.3 and Corollary E.7,

$$|\tilde{y}_i - \bar{y}_i| \leq \sqrt{\frac{2\log(2/\rho)}{m(\bar{p}_i + \ldots + \bar{p}_S)}}.$$

Therefore, with probability $1 - S\rho$,

$$
\begin{aligned}
&(\tilde{p} - \bar{p})^T h - \sum_i (\tilde{y}_i - \bar{y}_i)(h_i - \bar{H}_{i+1})(\bar{p}_i + \cdots + \bar{p}_S) \\
&= \sum_i (\tilde{y}_i - \bar{y}_i)(\tilde{H}_{i+1} - \bar{H}_{i+1})(\bar{p}_i + \cdots + \bar{p}_S) \\
&\geq -\sum_i \sqrt{\frac{2\log(2/\rho)}{m(\bar{p}_i + \ldots + \bar{p}_S)}} D\sqrt{\frac{2\log(2/\rho)}{m(\bar{p}_i + \ldots + \bar{p}_S)}}(\bar{p}_i + \cdots + \bar{p}_S) \\
&\geq -\frac{2SD\log(2/\rho)}{m}.
\end{aligned}
\tag{13}
$$

Then, applying Lemma A.4 (given $m\bar{p}_i \geq 6$) for $z_i = (h_i - \bar{H}_{i+1})(\bar{p}_i + \cdots + \bar{p}_S), i = 1,\ldots,S$, with probability $\Omega(1/S)$,

$$(\tilde{p} - \bar{p})^T h \geq \frac{1}{4}\sqrt{\sum_i z_i^2\bar{\sigma}_i^2} - \frac{2SD\log(2/\rho)}{m}.$$

Now, we observe

$$\sum_i z_i^2\bar{\sigma}_i^2 = (h_i - \bar{H}_{i+1})^2(\bar{p}_i + \cdots + \bar{p}_S)^2\bar{\sigma}_i^2 = \frac{\bar{c}_i^2\bar{p}_i(\bar{p}_i + \ldots, \bar{p}_S)}{m(\bar{p}_i + \ldots + \bar{p}_S) + 1},$$

to obtain

$$(\tilde{p} - \bar{p})^T h \geq \frac{1}{8}\sqrt{\sum_i \frac{\bar{\gamma}_i \bar{c}_i^2}{m}} - \frac{2SD\log(2/\rho)}{m}$$

where

$$\bar{\gamma}_i = \frac{\bar{p}_i(\bar{p}_{i+1} + \ldots + \bar{p}_S)}{(\bar{p}_i + \ldots + \bar{p}_S)}.$$

$\square$

## B  Optimism

In this section, we prove the following lemmas.

**Lemma 4.2.** *With probability $1 - \rho$, for every epoch $k$, the optimal gain $\tilde{\lambda}_k$ of the extended MDP $\tilde{\mathcal{M}}^k$ satisfies:*

$$\tilde{\lambda}_k \geq \lambda^* - O\left(D\log^2(T/\rho)\sqrt{\tfrac{SA}{T}}\right),$$

*where $\lambda^*$ the optimal gain of MDP $\mathcal{M}$ and $D$ is the diameter.*

*Proof.* Let $h^*$ be the bias vector for an optimal policy $\pi^*$ of MDP $\mathcal{M}$ (refer to Lemma 2.1 in the preliminaries section). Since $h^*$ is a fixed (though unknown) vector with $|h_i - h_j| \leq D$, we can apply Lemma 4.3 to obtain that with probability $1 - \rho$, for all $s, a$, there exists a sample vector $Q_{s,a}^{j,k}$ for some $j \in \{1, \ldots, \psi\}$ such that

$$(Q_{s,a}^{j,k})^T h^* \geq P_{s,a}^T h^* - \delta$$

where $\delta = O\left(D\log^2(T/\rho)\sqrt{\tfrac{SA}{T}}\right)$. Now, consider the policy $\pi$ for MDP $\tilde{\mathcal{M}}^k$ which for any $s$, takes action $a^j$, where $a = \pi^*(s)$, and $j$ is a sample satisfying above inequality. Note that $\pi$ is essentially $\pi^*$ but with a different transition probability model. Let $Q_\pi$ be the transition matrix for this policy, whose rows are formed by the vectors $Q_{s,\pi^*(s)}^{j,k}$, and $P_{\pi^*}$ be the transition matrix whose rows are formed by the vectors $P_{s,\pi^*(s)}$. Above implies

$$Q_\pi h^* \geq P_{\pi^*} h^* - \delta \mathbf{1}.$$

Let $Q_\pi^*$ denote the limiting matrix for Markov chain with transition matrix $Q_\pi$. Observe that $Q_\pi$ is aperiodic, recurrent and irreducible : it is aperiodic and irreducible because each entry of $Q_\pi$ being a sample from Dirichlet distribution is non-zero, and it is positive recurrent because in a finite irreducible Markov chain, all states are positive and recurrent. This implies that $Q_\pi^*$ is of the form $\mathbf{1}\mathbf{q}^{*T}$ where $\mathbf{q}^*$ is the stationary distribution of $Q_\pi$, and $\mathbf{1}$ is the vector of all 1s (refer to (A.6) in Puterman [2014]). Also, $Q_\pi^* Q_\pi = Q_\pi$, and $Q_\pi^* \mathbf{1} = \mathbf{1}$.

Therefore, the gain of policy $\pi$

$$\tilde{\lambda}(\pi)\mathbf{1} = (r_\pi^T \mathbf{q}^*)\mathbf{1} = Q_\pi^* r_\pi$$

where $r_\pi$ is the $S$ dimensional vector $[r_{s,\pi(s)}]_{s=1,\ldots,S}$. Now,

$$
\begin{aligned}
\tilde{\lambda}(\pi)\mathbf{1} - \lambda^*\mathbf{1} &= Q_\pi^* r_\pi - \lambda^*\mathbf{1} \\
&= Q_\pi^* r_\pi - \lambda^*(Q_\pi^* \mathbf{1}) &&\ldots \text{(using } Q_\pi^* \mathbf{1} = \mathbf{1}) \\
&= Q_\pi^*(r_\pi - \lambda^*\mathbf{1}) \\
&= Q_\pi^*(I - P_{\pi^*})h^* &&\ldots \text{(using (1))} \\
&= Q_\pi^*(Q_\pi - P_{\pi^*})h^* &&\ldots \text{(using } Q_\pi^* Q_\pi = Q_\pi^*) \\
&\geq -\delta\mathbf{1} &&\ldots \text{(using } (Q_\pi - P_{\pi^*})h^* \geq -\delta\mathbf{1}, Q_\pi^* \mathbf{1} = \mathbf{1}).
\end{aligned}
$$

Then, by optimality,

$$\tilde{\lambda}_k \geq \tilde{\lambda}(\pi) \geq \lambda^* - \delta.$$

$\square$

**Lemma 4.3.** (**Optimistic Sampling**) *Fix any vector $h \in \mathbb{R}^S$ such that $|h_i - h_{i'}| \leq D$ for any $i, i'$, and any epoch $k$. Then, for every $s, a$, with probability $1 - \frac{\rho}{SA}$ there exists at least one $j$ such that*

$$(Q_{s,a}^{j,k})^T h \geq P_{s,a}^T h - O\left(D \log^2(T/\rho)\sqrt{\frac{SA}{T}}\right).$$

*Proof.* For $s, a$ with $N_{s,a}^{\tau_k} \geq \eta$, $Q_{s,a}^{j,k}$ were generated using posterior sampling from Dirichlet distribution Dirichlet$(M_{s,a}^{\tau_k}(i), i = 1, \ldots, S)$. We use Proposition B.3 for optimism of a Dirichlet posterior sample. Let's verify the conditions applying for this proposition. We have $N_{s,a}^{\tau_k} \geq \eta = \sqrt{\frac{TS}{A}} + 12\omega S^2 \geq 12\omega S^2$. and $\omega = 720 \log(n/\rho)$.

Therefore, applying Proposition B.3, with probability $\Omega(1/S)$, the $j^{th}$ sample $Q_{s,a}^{j,k}$ satisfies the following kind of optimism:

$$(Q_{s,a}^{j,k})^T h \geq P_{s,a}^T h - O(\frac{DS \log^2(n/\rho)}{N_{s,a}^{\tau_k}}).$$

Substituting $N_{s,a}^{\tau_k} \geq \eta = \sqrt{\frac{TS}{A}} + 12\omega S^2$ we get that every $j$ satisfies the stated condition with probability $\Omega(1/S)$.

For $s, a$ with $N_{s,a}^{\tau_k} \leq \eta$, we used simple optimistic sampling. In Lemma B.1 we show for such $s, a$ the condition $(Q_{s,a}^{j,k})^T h \geq P_{s,a}^T h$ is satisfied by any $j$ with probability $1/2S$.

Therefore, given that the number of samples is $\psi = CS \log(SA/\rho)$ for some large enough constant $C$, for every $s, a$, with probability $1 - \frac{\rho}{SA}$, there exists at least one sample $Q_{s,a}^{j,k}$ satisfying the required condition.

$\square$

**Notations** We fix some notations for the rest of the section. Fix an epoch $k$, state and action pair $s, a$, sample $j$. In below, we denote $n = N_{s,a}^{\tau_k}$, $n_i = N_{s,a}^{\tau_k}(i), p_i = P_{s,a}(i), \hat{p}_i := \frac{n_i}{n}, \bar{p}_i = \frac{n_i + \omega}{n + \omega S}, \tilde{p}_i = Q_{s,a}^{j,k}(i)$, for $i \in \mathcal{S}$.

## B.1 Optimism for $n \leq \eta$ (**Simple Optimistic Sampling**)

When $n < \eta$, simple optimistic sampling is used, so that any sample vector $\tilde{p}$ was generated as follows: we let $p^- = [\hat{p} - (\sqrt{\frac{3\hat{p}_i \log(4S)}{n}} + \frac{3\log(4S)}{n})\mathbf{1}]^+$, and let $\mathbf{z}$ be a random vector picked uniformly at random from $\{\mathbf{1}_1, \ldots, \mathbf{1}_S\}$, and set

$$\tilde{p} = p^- + (1 - \sum_j p_j^-)\mathbf{z}.$$

We prove the following lemmas for this sample vector.

**Lemma B.1.** *For any fixed $h \in [0, D]^S$, we have*

$$\tilde{p}^T h \geq p^T h,$$

*with probability at least $\Omega(1/S)$.*

*Proof.* Define $\delta_i := \hat{p}_i - p_i$ (and hence $\sum_i \delta_i = 0$). By multiplicative Chernoff bounds (Fact 4), with probability $1 - \frac{1}{2S}$, $|\delta_i| \leq \sqrt{\frac{3\hat{p}_i \log(4S)}{n}} + \frac{3\log(4S)}{n}$. Also define $\Delta_i := \hat{p}_i - p_i^- = \min\left\{\sqrt{\frac{3\hat{p}_i \log(4S)}{n}} + \frac{3\log(4S)}{n}, \hat{p}_i\right\}$. Note that $\Delta_i \geq \delta_i$ and $\sum_i \Delta_i = \sum_i(\hat{p}_i - p_i^-) = 1 - \sum_i p_i^-$.

With probability $1/S$, $z = \mathbf{1}_i$ is picked such that $h_i = D$, and (by union bound over all $i$) with probability $1 - S\frac{1}{2S} = \frac{1}{2}$, $|\delta_i| \leq \sqrt{\frac{3\log(4S)}{n}} + \frac{3\log(4S)}{n}$ for every $i$. So with probability $1/2S$:

$$\begin{aligned}
\sum_i \tilde{p}_i h_i &= \sum_i p_i^- h_i + D(1 - \sum_j p_j^-) = \sum_i p_i^- h_i + D \sum_j \Delta_j \\
&= \sum_i (\hat{p}_i - \Delta_i) h_i + D\Delta_i = \sum_i \hat{p}_i h_i + (D - h_i)\Delta_i \\
&\geq \sum_i \hat{p}_i h_i + (D - h_i)\delta_i = \sum_i (\hat{p}_i - \delta_i) h_i + D\delta_i \\
&= \sum_i p_i h_i + D \sum_i \delta_i = \sum_i p_i h_i.
\end{aligned}$$

$\square$

Using the same technique as above, we can also prove the following "pessimism" for these samples, which will be used later, in bounding the diameter in Section D.

**Lemma B.2** (Pessimism). *When $n < \eta$, we have for any fixed $h \in [0, D]^S$*

$$\tilde{p}^T h \leq p^T h,$$

*with probability at least $\Omega(1/S)$.*

*Proof.* Define $\delta_i, \Delta_i$ as before. With probability $1/S$, $z = \mathbf{1}_i$ is picked such that $h_i = 0$, and again with probability $1 - S\frac{1}{2S} = \frac{1}{2}$, $|\delta_i| \leq \sqrt{\frac{3\log(4S)}{n}} + \frac{3\log(4S)}{n}$ for every $i$. So with probability $1/2S$:

$$\begin{aligned}
\sum_i \tilde{p}_i h_i &= \sum_i p_i^- h_i \\
&= \sum_i (\hat{p}_i - \Delta_i) h_i \\
&\leq \sum_i (\hat{p}_i - \delta_i) h_i \\
&= \sum_i p_i h_i.
\end{aligned}$$

$\square$

## B.2 Optimism for $n > \eta$ (Dirichlet posterior sampling)

When $n > \eta$, Dirichlet posterior sampling is used so that $\tilde{p}$ is a random vector distributed as Dirichlet$(m\bar{p}_1, \ldots, m\bar{p}_S)$, where $m = \frac{n+\omega S}{\kappa}$, $\bar{p} = \frac{n_i + \omega}{n + \omega S}$. We prove an optimism property for this sample vector. Following notations will be useful.

$$\gamma_i := \frac{p_i(p_{i+1} + \ldots + p_S)}{(p_i + \ldots + p_S)}, c_i := (h_i - H_{i+1}), H_{i+1} = \frac{1}{\sum_{j=i+1}^S p_j} \sum_{j=i+1}^S h_j p_j$$

$$\bar{\gamma}_i := \frac{\bar{p}_i(\bar{p}_{i+1} + \ldots + \bar{p}_S)}{(\bar{p}_i + \ldots + \bar{p}_S)}, \bar{c}_i := (h_i - \bar{H}_{i+1}), \bar{H}_{i+1} = \frac{1}{\sum_{j=i+1}^S \bar{p}_j} \sum_{j=i+1}^S h_j \bar{p}_j$$

where the states are indexed from 1 to $S$ such that $\bar{p}_1 \leq \cdots \leq \bar{p}_S$.

**Proposition B.3.** *Assuming $\omega = 720 \log(n/\rho) \geq 613 \log(2/\rho), n > 12\omega S^2, \kappa = 120 \log(n/\rho) = \frac{\omega}{6}$, then with probability $\Omega(1/S) - 8S\rho$,*

$$\tilde{p}^T h \geq p^T h - O(\frac{DS \log^2(n/\rho)}{n}).$$

*Proof.* The proof of this proposition involves showing that with probability $\Omega(1/S) - 8S\rho$, the random quantity $\tilde{p}^T h$ exceeds its mean $\bar{p}^T h$ enough to overcome the possible deviation of empirical estimate $\bar{p}^T h$ from the true value $p^T h$. This involves a Dirichlet anti-concentration bound (Proposition A.1 and Lemma B.4) to lower bound $\tilde{p}^T h$, and a concentration bound on empirical estimates $\hat{p}$ (Lemma C.3) to lower bound $\bar{p}^T h$ which by definition is close to $\hat{p}^T h$.

In Lemma B.4, we show that with probability $\Omega(1/S) - 7S\rho$,

$$(\tilde{p} - \bar{p})^T h \geq 0.188 \sqrt{\sum_{i<S} \frac{\gamma_i c_i^2}{m}} - O(\frac{DS\omega \log(n/\rho)}{n}).$$

Note that $m = \frac{n+\omega S}{\kappa}$ and so $\frac{n}{\kappa} < m < \frac{25n}{24\kappa}$ since $n > 12\omega S^2$. Then we have that

$$(\tilde{p} - \bar{p})^T h \geq 0.184 \sqrt{\kappa \sum_i \frac{\gamma_i c_i^2}{n}} - O(\frac{DS\omega \log(n/\rho)}{n}).$$

We can also calculate

$$|(\bar{p} - \hat{p})^T h| = |\sum_{i=1}^{S} h_i(\frac{n\hat{p}_i + \omega}{n + \omega S} - \frac{n\hat{p}_i}{n})| = |\sum_i h_i(\frac{\omega(1 - S\hat{p}_i)}{n + \omega S})| \leq \frac{\omega DS}{n + \omega S} \leq \frac{\omega DS}{n}.$$

Finally, from Lemma C.3 bounding the deviation of empirical estimates, we have that with probability $1 - \rho$,

$$|(\hat{p} - p)^T h| \leq 2\sqrt{\log(n/\rho)\sum_{i<S}\frac{\gamma_i c_i^2}{n}} + 2D\frac{\log(n/\rho)}{n}.$$

Hence putting everything together we have that with probability $\Omega(1/S) - 8S\rho$,

$$
\begin{aligned}
(\tilde{p} - p)^T h &= (\tilde{p} - \bar{p})^T h + (\bar{p} - \hat{p})^T h + (\hat{p} - p)^T h \\
&\geq (\tilde{p} - \bar{p})^T h - |(\bar{p} - \hat{p})^T h| - |(\hat{p} - p)^T h| \\
&\geq 0.184\sqrt{\kappa\sum_i \frac{\gamma_i c_i^2}{n}} - 2\sqrt{\log(n/\rho)\sum_{i<S}\frac{\gamma_i c_i^2}{n}} - O(\frac{DS\omega\log(n/\rho)}{n}) \\
&\geq -O(\frac{DS\log^2(n/\rho)}{n})
\end{aligned}
$$

where the last inequality follows with $\omega = 720\log(n/\rho)$ and $\kappa = 120\log(n/\rho)$.

$\square$

**Lemma B.4.** *Assume that $h \in [0, D]^S$, and $\omega \geq 613\log(2/\rho), n > 12\omega S^2, \kappa = \frac{\omega}{6}$, and an ordering of $i$ such that $\bar{p}_1 \leq \cdots \leq \bar{p}_S$. Then, with probability $\Omega(1/S) - 7S\rho$,*

$$(\tilde{p} - \bar{p})^T h \geq 0.188\sqrt{\sum_i \frac{\gamma_i c_i^2}{m}} - O(\frac{DS\omega\log(n/\rho)}{n}).$$

*Proof.* The proof is obtained by a modification to the proof of Proposition A.1, which proves a similar bound but in terms of $\bar{\gamma}_i$'s and $\bar{c}_i$'s.

In the proof of that proposition, we obtain (refer to Equation (13)), with probability $1 - S\rho$ (assuming $m\bar{p}_i \geq 6$),

$$
\begin{aligned}
(\tilde{p} - \bar{p})^T h &\geq \sum_i(\tilde{y}_i - \bar{y}_i)(h_i - \bar{H}_{i+1})(\bar{p}_i + \cdots + \bar{p}_S) - \frac{2DS\log(2/\rho)}{m} \\
&\geq \sum_i(\tilde{y}_i - \bar{y}_i)(h_i - \bar{H}_{i+1})(\bar{p}_i + \cdots + \bar{p}_S) - O(\frac{DS\omega\log(n/\rho)}{n})
\end{aligned}
$$

where $\tilde{y}_i := \frac{\tilde{p}_i}{\tilde{p}_i + \cdots + \tilde{p}_S}$, $\bar{y}_i := \frac{\bar{p}_i}{\bar{p}_i + \cdots + \bar{p}_S}$, $\tilde{H}_{i+1} = \frac{1}{\sum_{j=i+1}^{S} \tilde{p}_j} \sum_{j=i+1}^{S} h_j \tilde{p}_j$, $\bar{H}_{i+1} = \frac{1}{\sum_{j=i+1}^{S} \bar{p}_j} \sum_{j=i+1}^{S} h_j \bar{p}_j$. Now, breaking up the term in the summation and using Lemma B.7 to bound $|H_{i+1} - \bar{H}_{i+1}|(\bar{p}_i + \cdots + \bar{p}_S)$ (since we have by assumption that $\omega \geq 613 \log(2/\rho)$ and $n > 12\omega S^2$) and Lemma E.4 and Corollary E.7 to bound $|\tilde{y}_i - \bar{y}_i|$, we get that for every $i$, with probability $1 - 4S\rho$,

$$(\tilde{p} - \bar{p})^T h - \sum_i (\tilde{y}_i - \bar{y}_i)(h_i - H_{i+1})(\bar{p}_i + \cdots + \bar{p}_S) + O\left(\frac{DS\omega \log(n/\rho)}{m}\right)$$

$$\geq \sum_i (\tilde{y}_i - \bar{y}_i)(\bar{H}_{i+1} - H_{i+1})(\bar{p}_i + \cdots + \bar{p}_S)$$

$$\geq -\sum_i \sqrt{\frac{2\log(2/\rho)}{m(\bar{p}_i + \cdots + \bar{p}_S)}} \left(3D\sqrt{\log(n/\rho)\frac{(\bar{p}_i + \cdots + \bar{p}_S)}{n}} + 4\frac{(\omega S + \log(n/\rho))D}{n}\right)$$

$$(*) \quad \geq \quad -\frac{6DS\sqrt{\log(2/\rho)\log(n/\rho)}}{\sqrt{mn}} - \frac{4(\omega S + \log(n/\rho))D\sqrt{2\log(2/\rho)}}{n\sqrt{m}} \sum_i \frac{1}{\sqrt{(\bar{p}_i + \cdots + \bar{p}_S)}}.$$

Recall that $m = \frac{n + \omega S}{\kappa}$, so that for $n > S\omega$, $n \geq \frac{m\kappa}{2} = \frac{m\omega}{12} \geq m \log(2/\rho)$, and the first term of $(*)$ is at least:

$$-\frac{6DS\sqrt{\log(2/\rho)\log(n/\rho)}}{\sqrt{m^2 \log(2/\rho)}} = -\frac{6DS\sqrt{\log(n/\rho)}}{m} = -O\left(\frac{DS\omega \log(n/\rho)}{n}\right).$$

Then using Lemma B.5 and $m = (n + S\omega)/\kappa > 6n/\omega > 72S^2$, the second term in $(*)$ is at least:

$$-\frac{8S(\omega S + \log(n/\rho))D\sqrt{2\log(2/\rho)}}{n\sqrt{72S^2}} = -O\left(\frac{DS\omega \log(n/\rho)}{n}\right).$$

Then, applying Lemma A.4 (given $m\bar{p}_i \geq 6$) for $z_i = (h_i - H_{i+1})(\bar{p}_i + \cdots + \bar{p}_S)$, $i = 1, \ldots, S$, with probability $\Omega(1/S)$,

$$\sum_i (\tilde{y}_i - \bar{y}_i)z_i \geq \frac{1}{4}\sqrt{\sum_i \bar{\sigma}_i^2 z_i^2}.$$

We substitute this in the above, with the observation

$$\sum_i z_i^2 \bar{\sigma}_i^2 = \sum_i (h_i - H_{i+1})^2 (\bar{p}_i + \cdots + \bar{p}_S)^2 \bar{\sigma}_i^2 = \sum_i \frac{c_i^2 \bar{p}_i (\bar{p}_i + \ldots, \bar{p}_S)}{m(\bar{p}_i + \ldots + \bar{p}_S) + 1} \geq \sum_i \frac{6}{7}\frac{\bar{\gamma}_i c_i^2}{m}.$$

So far we have that with probability $\Omega(1/S) - 4S\rho$,

$$(\tilde{p} - \bar{p})^T h \geq \frac{\sqrt{6}}{4\sqrt{7}}\sqrt{\sum_i \frac{\bar{\gamma}_i c_i^2}{m}} - O\left(\frac{DS\omega \log(n/\rho)}{n}\right). \tag{14}$$

Finally, we use Lemma B.6 with $k = 14$ (this requires $\omega \geq 613 \log(2/\rho)$) to lower bound $\bar{\gamma}_i$ by $\frac{1}{1.51}\gamma_i - O(\frac{\omega S}{n})$ to get with probability $\Omega(1/S) - 7S\rho$,

$$(\tilde{p} - \bar{p})^T h \quad \geq \quad 0.188\sqrt{\sum_i \frac{\gamma_i c_i^2}{m}} - O\left(\frac{DS\omega \log(n/\rho)}{n}\right).$$

$\square$

**Lemma B.5.** *Let $x \in \mathbb{R}^n$ such that $0 \leq x_1 \leq \cdots \leq x_n \leq 1$ and $\sum_i x_i = 1$. Then*

$$\sum_{i=1}^{n} \frac{1}{\sqrt{x_i + \cdots x_n}} \leq 2n.$$

*Proof.* Define $f(y) := \frac{1}{\sqrt{x_y + \cdots + x_n}}$ for all $y = 1, \cdots, n$. We prove that $x^* := (\frac{1}{n}, \frac{1}{n}, \cdots, \frac{1}{n})$ achieves the maximum value. Consider any solution $x'$. Assume there exists some index pair $i, j$ with $i < j$ and some $\epsilon > 0$ such that $x'_i \neq x'_j$ and increasing $x'_i$ by $\epsilon$ and decreasing $x'_j$ by $\epsilon$ preserves the ordering of the indices. This strictly increases the objective, because $f(k)$ strictly increases for all $i < k \leq j$ and remains unchanged otherwise, and hence $x'$ is not an optimal solution. The only case where no such index pair $(i, j)$ exists is when every $x_i$ is equal- this is precisely the solution $x^*$. Since $\sum_i f(i)$ is a continuous functions over a compact set, it has a maximum, which therefore must be attained at $x^*$.

This means

$$\sum_{i=1}^n \frac{1}{\sqrt{x_i + \cdots x_n}} \leq \sum_{i=1}^n \frac{1}{\sqrt{x_i^* + \cdots + x_n^*}} = \sum_{i=1}^n \sqrt{\frac{n}{i}} \leq \sqrt{n} \int_{i=0}^n \frac{1}{\sqrt{i}} di = 2n.$$

$\square$

**Lemma B.6.** *Let $A = 3\log(\frac{2}{\rho})$ and $\omega \geq \frac{25}{24} k^2 A$. Also let $n > 12\omega S^2$. Then for any group $\mathcal{G}$ of indices, with probability $1 - \rho$,*

$$(1 - \frac{1}{k}) \sum_{i \in \mathcal{G}} \bar{p}_i - \frac{2\omega S}{n} \leq \sum_{i \in \mathcal{G}} p_i \leq (1 + \frac{1}{k}) \sum_{i \in \mathcal{G}} \bar{p}_i + \frac{2\omega S}{n}.$$

*If in the definition of $\bar{\gamma}_i$, we use an ordering of $i$ such that $\bar{p}_S \geq \frac{1}{S}$ (e.g., if $\max \bar{p}_i$ is the last in the ordering), then for all $i$, with probability $1 - 3\rho$,*

$$\gamma_i \leq \frac{(1 + \frac{1}{k})^2}{1 - \frac{1}{k} - \frac{1}{6}} \bar{\gamma}_i + \frac{2(1 + \frac{1}{k} + \frac{1}{6}) \omega S}{1 - \frac{1}{k} - \frac{1}{6}} \frac{\omega S}{n}.$$

*Proof.* By multiplicative Chernoff-Hoeffding bounds (Fact 4), with probability $1 - \rho$,

$$|\sum_i p_i - \sum_i \hat{p}_i| \leq \sqrt{\frac{A \sum_i \hat{p}_i}{n}} + \frac{A}{n}$$

where $A = 3\log(\frac{2}{\rho})$ so that using $|\sum_i \bar{p}_i - \sum_i \hat{p}_i| \leq \frac{\omega S}{n}$,

$$|\sum_i p_i - \sum_i \bar{p}_i| \leq \sqrt{\frac{\sum_i \bar{p}_i A}{n}} + \frac{\sqrt{A\omega S}}{n} + \frac{A}{n} + \frac{\omega S}{n} \leq \sqrt{\frac{\sum_i \bar{p}_i A}{n}} + \frac{2\omega S}{n}.$$

Now, for $n > 12\omega S^2$, $n\bar{p}_i = n\frac{n\hat{p}_i + \omega}{n + \omega S} \geq \frac{n\omega}{n + \omega S} \geq \frac{24\omega}{25} \geq k^2 A$.

$$|\sum_i p_i - \sum_i \bar{p}_i| \leq \sum_i \bar{p}_i \sqrt{\frac{A}{n \sum_i \bar{p}_i}} + \frac{2\omega S}{n} \leq \sum_i \bar{p}_i \sqrt{\frac{A}{k^2 A}} + \frac{2\omega S}{n} \leq \frac{1}{k} \sum_i \bar{p}_i + \frac{2\omega S}{n}$$

so that

$$\sum_i p_i \leq (1 + \frac{1}{k}) \sum_i \bar{p}_i + \frac{2\omega S}{n}, \quad \sum_i p_i \geq (1 - \frac{1}{k}) \sum_i \bar{p}_i - \frac{2\omega S}{n}.$$

For the second statement of the lemma, using what we just proved, we have that with probability $1 - 3\rho$,

$$\gamma_i = \frac{p_i(p_{i+1} + \cdots + p_S)}{p_i + \cdots + p_S} \leq \frac{(1 + \frac{1}{k})^2 \bar{p}_i(\bar{p}_{i+1} + \cdots + \bar{p}_S) + \frac{2(1 + \frac{1}{k})\omega S(\bar{p}_i + \cdots + \bar{p}_S)}{n} + \frac{4\omega^2 S^2}{n^2}}{(1 - \frac{1}{k})(\bar{p}_i + \cdots + \bar{p}_S) - \frac{2\omega S}{n}}.$$

Now, if indices $i$ are ordered such that $\bar{p}_S \geq \frac{1}{S}$, then $\bar{p}_i + \cdots + \bar{p}_S \geq \frac{1}{S}$ for all $i$. Also, if $n > 12\omega S^2$, we have the following bound on the denominator in above: $(1 - \frac{1}{k})(\bar{p}_i + \cdots + \bar{p}_S) - \frac{2\omega S}{n} \geq (1 - \frac{1}{k} - \frac{1}{6})(\bar{p}_i + \cdots + \bar{p}_S)$, so that from above

$$\gamma_i \leq \frac{(1 + \frac{1}{k})^2}{1 - \frac{1}{k} - \frac{1}{6}} \bar{\gamma}_i + \frac{2(1 + \frac{1}{k} + \frac{1}{6}) \omega S}{1 - \frac{1}{k} - \frac{1}{6}} \frac{\omega S}{n}.$$

$\square$

**Lemma B.7.** *For any* **fixed** $h \in \mathbb{R}^S$, *and $i$, let* $\hat{H}_i = \frac{1}{\sum_{j=i}^{S} \hat{p}_j} \sum_{j=i}^{S} h_j \hat{p}_j$, $H_i = \frac{1}{\sum_{j=i}^{S} p_j} \sum_{j=i}^{S} h_j p_j$, $\bar{H}_i = \frac{1}{\sum_{j=i}^{S} \bar{p}_j} \sum_{j=i}^{S} h_j \bar{p}_j$. *Then if $n \geq 96$, with probability $1 - \rho$,*

$$|(\bar{H}_i - H_i)(\bar{p}_i + \ldots + \bar{p}_S)| \leq 2D\sqrt{\log(n/\rho)\frac{(p_i + \cdots + p_S)}{n}} + 3\frac{(\omega S + \log(n/\rho))D}{n}.$$

*Moreover, if we also assume that $\omega \geq 30\log(2/\rho)$ and $n > 12\omega S^2$, then with probability $1 - 2\rho$,*

$$|(\bar{H}_i - H_i)(\bar{p}_i + \ldots + \bar{p}_S)| \leq 3D\sqrt{\log(n/\rho)\frac{(\bar{p}_i + \cdots + \bar{p}_S)}{n}} + 4\frac{(\omega S + \log(n/\rho))D}{n}.$$

*Proof.* For every $t, k \geq i$, define

$$Z_{t,k} = \left(h_k \mathbb{1}(s_t = k) - h_k \frac{p_k}{p_i + \cdots + p_S} \cdot \mathbb{1}(s_t \in \{i, \ldots, S\})\right) \mathbb{1}(s_{t-1} = s, a_{t-1} = a),$$

$$Z_t = \sum_{k \geq i} Z_{t,k}.$$

Then,

$$\frac{\sum_{t=1}^{\tau} Z_t}{n} = \sum_{k \geq i} h_k \hat{p}_k - \sum_{k \geq i} h_k \frac{p_k}{p_i + \cdots + p_S} \cdot (\hat{p}_i + \ldots + \hat{p}_S) = (\hat{H}_i - H_i)(\hat{p}_i + \ldots + \hat{p}_S)$$

where we used Fact 1 for the last equality. Now, $E[Z_t | s_{t-1}, a_{t-1}] = \sum_{k \geq i} E[Z_{t,k} | s_{t-1}, a_{t-1}] = 0$. Also, we observe that for any $t$, $Z_{t,k}$ and $Z_{t,j}$ for any $k \neq j$ are negatively correlated given the current state and action:

$$
\begin{aligned}
\mathbb{E}[Z_{t,k} Z_{t,j} | s_{t-1}, a_{t-1}] &= h_k h_j \mathbb{E}[\mathbb{1}(s_t = k)\mathbb{1}(s_t = j) - \mathbb{1}(s_t = j)\frac{p_k}{p_i + \cdots + p_S} \cdot \mathbb{1}(s_t \in \{i, \ldots, S\}) \\
&\quad - \mathbb{1}(s_t = k)\frac{p_j}{p_i + \cdots + p_S} \cdot \mathbb{1}(s_t \in \{i, \ldots, S\}) \\
&\quad + \frac{p_j p_k}{(p_i + \cdots + p_S)^2} \cdot \mathbb{1}(s_t \in \{i, \ldots, S\})] \\
&= h_k h_j \mathbb{E}[-\frac{2p_j p_k}{p_i + \cdots + p_S} + \frac{p_k p_j}{(p_i + \cdots + p_S)^2} \cdot \mathbb{1}(s_t \in \{i, \ldots, S\})] \\
&= h_k h_j \mathbb{E}[-\frac{p_j p_i}{p_i + \cdots + p_S}] \\
&\leq 0.
\end{aligned}
$$

And,

$$
\begin{aligned}
\mathbb{E}[\sum_{t=1}^{\tau} Z_{t,k}^2 | s_{t-1} = s, a_{t-1} = a] &= h_k^2 \sum_{\tau=1}^{t} \mathbb{1}(s_{t-1} = s, a_{t-1} = a)\left(p_k - \frac{p_k^2}{(p_i + \cdots + p_S)^2}(p_i + \cdots + p_S)\right) \\
&= h_k^2 \sum_{t=1}^{\tau} \mathbb{1}(s_{t-1} = s, a_{t-1} = a)\frac{p_k(\sum_{j \geq i, j \neq k} p_j)}{p_i + \cdots + p_S} \\
&= nh_k^2 \frac{p_k(\sum_{j \geq i, j \neq k} p_j)}{p_i + \cdots + p_S} \\
&\leq nD^2 p_k.
\end{aligned}
$$

Therefore,

$$\sum_{t=1}^{\tau} E[Z_t^2 | s_{t-1}, a_{t-1}] \leq \sum_{t=1}^{\tau} \sum_{k \geq i} \mathbb{E}[Z_{t,k}^2 | s_{t-1}, a_{t-1}] \leq nD^2(p_i + \cdots + p_S).$$

Then, applying Bernstein's inequality (refer to Corollary E.1) to bound $|\sum_{t=1}^{\tau} Z_t|$, we get the following bound on $\frac{1}{n} \sum_{t=1}^{\tau} Z_t = (\hat{H}_i - H_i)(\hat{p}_i + \ldots + \hat{p}_S)$ with probability $1 - \rho$:

$$|(\hat{H}_i - H_i)(\hat{p}_i + \ldots + \hat{p}_S)| = |\frac{1}{n} \sum_{t=1}^{\tau} Z_t| \leq 2D\sqrt{\log(n/\rho)\frac{(p_i + \cdots + p_S)}{n}} + 3D\frac{\log(n/\rho)}{n}.$$

Also,

$$|\hat{H}_i - \bar{H}_i| = |\sum_k \frac{\hat{p}_k}{\hat{p}_i + \cdots + \hat{p}_S} h_k - \frac{\bar{p}_k}{\bar{p}_i + \cdots + \bar{p}_S} h_k| \le \frac{\omega S D}{n(\hat{p}_i + \cdots + \hat{p}_S)},$$

Combining,

$$|(\bar{H}_i - H_i)(\hat{p}_i + \ldots + \hat{p}_S)| \le 2D\sqrt{\log(n/\rho)\frac{(p_i + \cdots + p_S)}{n}} + 3D\frac{\log(n/\rho)}{n} + \frac{\omega S D}{n}.$$

Replacing $\hat{p}_i$ by $\bar{p}_i$,

$$|(\bar{H}_i - H_i)(\bar{p}_i + \ldots + \bar{p}_S)| \le 2D\sqrt{\log(n/\rho)\frac{(p_i + \cdots + p_S)}{n}} + 3\frac{(\omega S + \log(n/\rho))D}{n}$$

with probability $1 - \rho$.

Now, if we also have that $\omega \ge 30\log(2/\rho)$ and $n > 12\omega S^2$, using lemma B.6 with $k = 3$ to replace $p_i$ by $\bar{p}_i$, with probability $1 - 2\rho$,

$$|(\bar{H}_i - H_i)(\bar{p}_i + \ldots + \bar{p}_S)| \le 3D\sqrt{\log(n/\rho)\frac{(\bar{p}_i + \cdots + \bar{p}_S)}{n}} + 4\frac{(\omega S + \log(n/\rho))D}{n}.$$

$\square$

## C  Deviation bounds

**Lemma 4.4.** (**Deviation bound**) *With probability* $1 - \rho$, *for all epochs* $k$, *sample* $j$, *all* $s, a$

$$\max_{h \in [0,2D]^S} (Q_{s,a}^{j,k} - P_{s,a})^T h \le \begin{cases} O\left(D\sqrt{\frac{\log(SAT/\rho)}{N_{s,a}^{\tau_k}}} + D\frac{S\log(SAT/\rho)}{N_{s,a}^{\tau_k}}\right), & N_{s,a}^{\tau_k} > \eta \\ O\left(D\sqrt{\frac{S\log(SAT/\rho)}{N_{s,a}^{\tau_k}}} + D\frac{S\log(S)}{N_{s,a}^{\tau_k}}\right), & N_{s,a}^{\tau_k} \le \eta \end{cases}$$

*Proof.* For $n > \eta$, express the above as

$$\max_{h \in [0,2D]^S} (Q_{s,a}^{j,k} - P_{s,a})^T h \le \max_{h \in [0,2D]^S} (Q_{s,a}^{j,k} - \bar{P}_{s,a})^T h + (\bar{P}_{s,a} - \hat{P}_{s,a})^T h + (\hat{P}_{s,a} - P_{s,a})^T h$$

where $\bar{P}_{s,a} = \frac{M_{s,a}^{\tau_k}(i)}{M_{s,a}^{\tau_k}} = \frac{N_{s,a}^{\tau_k}(i) + \omega}{N_{s,a}^{\tau_k} + \omega S}$ is the mean of Dirichlet$(\mathbf{M}_{s,a}^{\tau_k})$ distribution used to sample $Q^{j,k}$, and $\hat{P}_{s,a} = \frac{N_{s,a}^{\tau_k}(i)}{N_{s,a}^{\tau_k}}$. Now,

$$\max_{h \in [0,2D]^S} (\bar{P}_{s,a} - \hat{P}_{s,a})^T h \le \frac{2\omega S D}{N_{s,a}^{\tau_k}}.$$

And, to bound the first and the last terms in above, we use Lemma C.1 and Lemma C.2 with union bound for all $S, A, \psi, k$, to get the lemma statement for $n > \eta$.

For $n < \eta$, we use Lemma C.4 with a union bound for $S, A, \psi, k$, we get the lemma statement. $\square$

### C.1  Dirichlet concentration

A similar result as the lemma below for concentration of Dirichlet random vectors was proven in Osband and Van Roy [2016]. We include (an expanded version of) the proof for completeness.

**Lemma C.1** (Osband and Van Roy [2016]). *Let* $\tilde{p} \sim Dirichlet(m\bar{p})$. *Let*

$$Z := \max_{v \in [0,D]^S} (\tilde{p} - \bar{p})^T v.$$

*Then,* $Z \le D\sqrt{\frac{2\log(2/\rho)}{m}}$, *with probability* $1 - \rho$.

*Proof.* Define disjoint events $\mathcal{E}_v, v \in [0, D]^S$ in the sample space of $Z$ as

$$\mathcal{E}_v = \{Z : Z = \max_{w \in [0,D]^S} (\tilde{p} - \bar{p})^T w = (\tilde{p} - \bar{p})^T v\}.$$

Let $f(v)$ be the probability of event $\mathcal{E}_v$. (Here, ties are broken in arbitrary but fixed manner to assign each $Z$ to one of the $\mathcal{E}_v$ so that $\mathcal{E}_v$ are disjoint and $f(v)$ integrate to 1).

Now, define a random variable $Y$ distributed as follows: $Y = Y_v - E[Y_v]$ with probability $f(v)$, where $Y_v$s are Beta variables distributed as $Y_v \sim Beta(m\frac{1}{D}\bar{p}^T v, m(1 - \frac{1}{D}\bar{p}^T v))$. We show that $Y$ is stochastically optimistic compared to $Z$.

We couple $Y$ and $Z$ as follows: when $Z \in \mathcal{E}_v$, which is with probability $f(v)$, we set $Y$ is $Y_v$. By definition, under this event, $Z = (\tilde{p} - \bar{p})^T v$. By Dirichlet-Beta optimism (Lemma E.5), for any $v$, $DY_v$ is stochastically optimistic compared to $\tilde{p}^T v$. Now, since they have the same mean, from equivalence condition for stochastic optimism (Condition 3 in Lemma 3 of Osband et al. [2014])

$$\mathbb{E}[DY_v - \tilde{p}^T v | \tilde{p}^T v] = 0$$

for all values of $v, \tilde{p}^T v$. Since we coupled $Y$ and $Z$ so that $Y$ is $Y_v - \mathbb{E}[Y_v]$ when $Z \in \mathcal{E}_v$, we can derive that for any $v$, and $z \in \mathcal{E}_v$,

$$
\begin{aligned}
\mathbb{E}[DY - Z|Z = z : z \in \mathcal{E}_v] &= \mathbb{E}[DY_v - D\mathbb{E}[Y_v] - Z \mid Z = z : z \in \mathcal{E}_v] \\
&= \mathbb{E}[DY_v - D\mathbb{E}[Y_v] - (\tilde{p} - \bar{p})^T v \mid (\tilde{p} - \bar{p})^T v] \\
&= \mathbb{E}[DY_v - \tilde{p}^T v \mid \tilde{p}^T v] = 0.
\end{aligned}
$$

This is true for all $z$, since every $z \in \mathcal{E}_v$ for some $v$, thus proving

$$DY \succeq_{so} Z.$$

Let $X$ be distributed as Gaussian with mean 0 and variance $\frac{1}{m}$. By Gaussian-Beta stochastic optimism $X \succeq_{so} Y_v - \mathbb{E}[Y_v]$, which implies for any convex increasing $u(\cdot)$,

$$\mathbb{E}[u(Y)] = \int_v \mathbb{E}[u(Y_v - \mathbb{E}[Y_v])]f(v) \leq \int_v \mathbb{E}[u(X)]f(v) = \mathbb{E}[u(X)]$$

so that $X \succeq_{so} Y$, and

$$X \succeq_{so} Y \succeq_{so} \frac{1}{D} Z.$$

Therefore, we can use Corollary E.7 to bound $Z$ by $D\sqrt{\frac{2\log(2/\rho)}{m}}$ with probability $1 - \rho$. $\qquad\square$

### C.2 Concentration of average of independent multinoulli trials

Below we study concentration properties of vector $\hat{p}$ defined as the average of $n$ independent multinoulli trials with parameter $p \in \Delta^S$, i.e., $\hat{p} = \sum_{j=1}^n \mathbf{x}_j$, where $\mathbf{x}_j$s are iid random vectors, with $x_{ij} = 1$ with probability $p_i$.

**Lemma C.2.** *Let $\hat{p}$ be the average of $n$ independent multinoulli trials with parameter $p$. Let*

$$Z := \max_{v \in [0,D]^S} (\hat{p} - p)^T v.$$

*Then, $Z \leq D\sqrt{\frac{2\log(1/\rho)}{n}}$, with probability $1 - \rho$.*

*Proof.* Define disjoint events $\mathcal{E}_v, v \in [0, D]^S$ in the sample space of $Z$ as

$$\mathcal{E}_v = \{Z : Z = \max_{w \in [0,D]^S} (\hat{p} - p)^T w = (\hat{p} - p)^T v\}.$$

Let $f(v)$ be the probability of event $\mathcal{E}_v$. (Here, ties are broken in arbitrary but fixed manner to assign each $Z$ to one of the $\mathcal{E}_v$ so that $\mathcal{E}_v$ are disjoint and $f(v)$ integrate to 1).

Now, define a random variable $Y$ distributed as follows: $Y = Y_v - E[Y_v]$ with probability $f(v)$, where $Y_v$s are independent Binomial variables distributed as $Y_v \sim \frac{1}{n}Binomial(n, \frac{1}{D}p^T v)$. We show that $Y$ is stochastically optimistic compared to $Z$.

We couple $Y$ and $Z$ as follows: when $Z \in \mathcal{E}_v$, which is with probability $f(v)$, we set $Y$ is $Y_v$. By definition, under this event, $Z = (\hat{p} - p)^T v$. By Multinomial-Binomial optimism (Lemma E.8 and Corollary E.9), for any $v$, $DY_v$ is stochastically optimistic compared to $\hat{p}^T v$. Now, since they have the same mean, from equivalence condition for stochastic optimism (Condition 3 in Lemma 3 of Osband et al. [2014])

$$\mathbb{E}[DY_v - \hat{p}^T v | \hat{p}^T v] = 0$$

for all values of $v, \tilde{p}^T v$. Since we coupled $Y$ and $Z$ so that $Y$ is $Y_v - \mathbb{E}[Y_v]$ when $Z \in \mathcal{E}_v$, we can derive that for any $v$, and $z \in \mathcal{E}_v$,

$$
\begin{aligned}
\mathbb{E}[DY - Z | Z = z : z \in \mathcal{E}_v] &= \mathbb{E}[DY_v - D\mathbb{E}[Y_v] - Z \mid Z = z : z \in \mathcal{E}_v] \\
&= \mathbb{E}[DY_v - D\mathbb{E}[Y_v] - (\hat{p} - p)^T v \mid (\hat{p} - p)^T v] \\
&= \mathbb{E}[DY_v - \hat{p}^T v \mid \hat{p}^T v] = 0.
\end{aligned}
$$

This is true for all $z$, since every $z \in \mathcal{E}_v$ for some $v$, thus proving

$$DY \succeq_{so} Z.$$

Next, we bound $Z$ using the stochastic optimism. First, let us express the distribution of $Y$ in a more convenient way. Let $\mu_v = \frac{1}{D} p^T v, \mu = \int_v f(v) \mu_v$. Define

$$X = \sum_{j=1}^{n} X^j$$

where $X^j$s are iid random variables, distributed as follows: $X^j$ takes value $1 - \mu_v$ with probability $f(v)\mu_v$ and $-\mu_v$ w.p. $f(v)(1 - \mu_v)$, for $v \in [0, D]^d$. Therefore, $\mathbb{E}[X^j] = \int_v (1 - \mu_v) f(v)\mu_v - \mu_v f(v)(1 - \mu_v) = 0$, and $X^j \in [-1, 1]$. We show that $X$ and $Y$ have the same distribution.

Since each $Y_v$ is Binomial$(n, \mu_v)$, we can write it as $Y_v = \sum_{j=1}^{n} Y_v^j$ where $Y_v^j$ are independent Bernoulli$(\mu_v)$ random variables. Define a random variable $\tilde{v}$ which is $v$ with probability $f(v)$. Then, since $Y$ is $Y_v - \mu_v$ w.p. $f(v)$,

$$Y \sim \int_v (Y_v - \mu_v) \mathbb{1}(\tilde{v} = v) = \frac{1}{n} \sum_j \int_v (1 - \mu_v) \mathbb{1}(\tilde{v} = v, Y_v^j = 1) - \mu_v \mathbb{1}(\tilde{v} = v, Y_v^j = 0) \sim \frac{1}{n} \sum_j X^j.$$

Therefore,

$$X \sim Y \succeq_{so} \frac{1}{D} Z$$

where $X = \frac{1}{n} \sum_{j=1}^{n} X^j$, is the sum of $n$ mean 0, bounded $[-1, 1]$, iid random variables. By Hoeffding's lemma, for any $s \in \mathbb{R}$

$$\mathbb{E}[e^{sX^j}] \le e^{\frac{s^2}{2}}, \text{ so that, } \mathbb{E}[e^{snX}] \le e^{\frac{ns^2}{2}}.$$

Using stochastic optimism $\mathbb{E}[u(Z/d)] \le E[u(Y)] = E[u(X)]$ for all convex increasing $u(\cdot)$, therefore for $s > 0$,

$$P(n\frac{Z}{D} > nt) \le \frac{\mathbb{E}[e^{sn\frac{Z}{D}}]}{e^{snt}} \le \frac{\mathbb{E}[e^{snX}]}{e^{snt}} \le e^{\frac{ns^2}{2} - snt}.$$

Choosing $s = t = \sqrt{\frac{2 \log(1/\rho)}{n}}$,

$$P(\frac{Z}{D} > \sqrt{\frac{\log(1/\rho)}{n}}) \le e^{\frac{-t^2}{2}} < \rho.$$

$\square$

**Lemma C.3.** *Let $\hat{p} \in \Delta^S$ be the average $n$ independent multinoulli trials with parameter $p \in \Delta^S$. Then, for any **fixed** $h \in [0, D]^S$ and $n \ge 96$, with probability $1 - \rho$,*

$$|(\hat{p} - p)^T h| \le 2\sqrt{\log(n/\rho) \sum_{i < S} \frac{\gamma_i c_i^2}{n}} + 3D \frac{\log(2/\rho)}{n},$$

*where $\gamma_i = \frac{p_i(p_{i+1} + \cdots + p_S)}{(p_i + \cdots + p_S)}$, $c_i = h_i - H_{i+1}$, $H_{i+1} = \frac{1}{\sum_{j=i+1}^{S} p_j} \sum_{j=i+1}^{S} h_j p_j$.*

*Proof.* For every $t, i$, define

$$Z_{t,i} = \left( c_i \mathbb{1}(s_t = i) - c_i \frac{p_i}{p_i + \cdots + p_S} \cdot \mathbb{1}(s_t \in \{i, \ldots, S\}) \right) \mathbb{1}(s_{t-1} = s, a_{t-1} = a),$$

$$Z_t = \sum_i Z_{t,i}.$$

Then,

$$\frac{\sum_{t=1}^{\tau} Z_t}{n} = \sum_i c_i \hat{p}_i - \sum_i \frac{c_i p_i}{p_i + \cdots + p_S} \cdot (\hat{p}_i + \ldots + \hat{p}_S) = \sum_{i=1}^{S-1} (\hat{y}_i - y_i)(\hat{p}_i + \ldots + \hat{p}_S) c_i = (\hat{p} - p)^T h$$

where we used Fact 1 for the last equality. Now, $E[Z_t | s_{t-1}, a_{t-1}] = \sum_i E[Z_{t,i} | s_{t-1}, a_{t-1}] = 0$. Also, we observe that for any $t$, $Z_{t,i}$ and $Z_{t,j}$ for any $i \neq j$ are independent given the current state and action: (assume $j > i$ w.l.o.g.)

$$
\begin{aligned}
\mathbb{E}[Z_{t,i} Z_{t,j} | s_{t-1}, a_{t-1}] &= c_i c_j \mathbb{E}[\mathbb{1}(s_t = i) \mathbb{1}(s_t = j) - \mathbb{1}(s_t = j) \frac{p_i}{p_i + \cdots + p_S} \cdot \mathbb{1}(s_t \in \{i, \ldots, S\}) \\
&\quad - \mathbb{1}(s_t = i) \frac{p_j}{p_j + \cdots + p_S} \cdot \mathbb{1}(s_t \in \{j, \ldots, S\}) \\
&\quad + \frac{p_j p_i}{(p_j + \cdots + p_S)(p_i + \cdots + p_S)} \cdot \mathbb{1}(s_t \in \{j, \ldots, S\})] \\
&= c_i c_j \mathbb{E}[-\mathbb{1}(s_t = j) \frac{p_i}{p_i + \cdots + p_S} \\
&\quad + \frac{p_j p_i}{(p_j + \cdots + p_S)(p_i + \cdots + p_S)} \cdot \mathbb{1}(s_t \in \{j, \ldots, S\})] \\
&= c_i c_j \mathbb{E}[-\frac{p_j p_i}{p_i + \cdots + p_S} + \frac{p_j p_i}{(p_i + \cdots + p_S)}] \\
&= 0.
\end{aligned}
$$

Therefore,

$$\sum_{t=1}^{\tau} E[Z_t^2 | s_{t-1}, a_{t-1}] = \sum_{t=1}^{\tau} \sum_i c_i^2 \mathbb{E}[Z_{t,i}^2 | s_{t-1}, a_{t-1}] = \sum_i c_i^2 n \gamma_i,$$

where the last equality is obtained using the following derivation:

$$
\begin{aligned}
\mathbb{E}[\sum_{t=1}^{\tau} Z_{t,i}^2 | s_{t-1} = s, a_{t-1} = a] &= \sum_{t=1}^{\tau} \mathbb{1}(s_{t-1} = s, a_{t-1} = a) \left( p_i - \frac{p_i^2}{(p_i + \cdots + p_S)^2}(p_i + \cdots + p_S) \right) \\
&= \sum_{t=1}^{\tau} \mathbb{1}(s_{t-1} = s, a_{t-1} = a) \frac{p_i(p_{i+1} + \cdots + p_S)}{p_i + \cdots + p_S} \\
&= n \frac{p_i(p_{i+1} + \cdots + p_S)}{p_i + \cdots + p_S} = n \gamma_i.
\end{aligned}
$$

Then, applying Bernstein's inequality (refer to Corollary E.1) to bound $|\sum_{t=1}^{\tau} Z_t|$, we get the desired bound on $(p - \hat{p})^T h = \frac{1}{n} \sum_{t=1}^{\tau} Z_t$. $\qquad \square$

## C.3 Concentration of simple optimistic samples

**Lemma C.4.** *Let* $\tilde{p} = p^- + (1 - \sum_{i=1}^{S} p_i^-) \mathbf{z}$ *where* $\mathbf{z}$ *be a random vector picked uniformly at random from* $\{\mathbf{1}_1, \ldots, \mathbf{1}_S\}$, *and* $p^- = \hat{p} - \mathbf{\Delta}$, $\Delta_i = \min \left\{ \sqrt{\frac{3 \hat{p}_i \log(4S)}{n}} + \frac{3 \log(4S)}{n}, \hat{p}_i \right\}$, *then with probability at least* $1 - \rho$, *for any D, we have*

$$\max_{h \in [0, D]^S} (\tilde{p}^T h - p^T h) \leq O(D \sqrt{\frac{S \log(nS/\rho)}{n}} + \frac{DS \log(2S)}{n}).$$

*Proof.* By definition of $\tilde{p}$ and using Lemma C.2, with probability $1 - \rho$,

$$
\begin{aligned}
\max_{h \in [0,D]^S} (\tilde{p}^T h - p^T h) &\leq (\hat{p}^T h - p^T h) + D \sum_i \sqrt{\frac{3\hat{p}_i \log(4S)}{n}} + \sum_i \frac{3D \log(4S)}{n} \\
&\leq 2D\sqrt{\frac{2\log(1/\rho)}{n}} + D\sqrt{S\frac{3\log(4S)}{n}} + \frac{DS\log(4S)}{n} \\
&= O(D\sqrt{\frac{S\log(4S/\rho)}{n}} + \frac{DS\log(4S)}{n}).
\end{aligned}
$$

$\square$

## D   Diameter of the extended MDP $\tilde{\mathcal{M}}^k$

**Lemma 4.1.** *Assume $T \geq CDA \log^2(T/\rho)$ for a large enough constant $C$. Then, with probability $1 - \rho$, for every epoch $k$, the diameter of MDP $\tilde{\mathcal{M}}^k$ is bounded by $2D$.*

*Proof.* Using Lemma D.2, along with Lemma D.1 for $h = E^s$, we obtain that the diameter of $\tilde{\mathcal{M}}^k$ is bounded by $D/(1 - \delta)$ for $\delta = O(D\sqrt{\frac{\log(1/\rho)}{\eta}} + D\frac{\log(T/\rho)}{\eta})$, where $\eta = \sqrt{\frac{TS}{A}}$. Therefore, if $T \geq CDA \log^2(T/\rho)$, then $\eta \geq CDS \log(T/\rho) \geq CD^2 \log(1/\rho)$, making $\delta \leq 1/2$ for some large enough constant $C$. $\square$

**Lemma D.1.** *For every $k$, and any fixed $h \in [0, D]^S$, with probability $1 - \rho$, there exists a sample vector $Q_{s,a}^{j,k}$ such that*

$$
Q_{s,a}^{j,k} \cdot h \leq P_{s,a} \cdot h + O(D\sqrt{\frac{\log(1/\rho)}{\eta}} + DS\frac{\log(T/\rho)}{\eta}).
$$

*Proof.* First consider $s, a$ with $N_{s,a}^{\tau_k} \geq \eta$. For such $s, a$ posterior sampling is used, and by Lemmas C.1 and C.2,

$$
Q_{s,a}^{j,k} \cdot h \leq P_{s,a} \cdot h + O(D\sqrt{\frac{\log(1/\rho)}{N_{s,a}^{\tau_k}}} + D\frac{\omega S}{N_{s,a}^{\tau_k}}) \leq P_{s,a} \cdot h + O(D\sqrt{\frac{\log(1/\rho)}{\eta}} + DS\frac{\log(T/\rho)}{\eta}).
$$

For $s, a$ with $N_{s,a}^{\tau_k} \leq \eta$, we use a simple optimistic sampling. In Lemma B.2, we prove that under such sampling $Q_{s,a}^{j,k} \cdot h \leq P_{s,a} \cdot h$ with probability $1/2S$ for every sample $j$. Then, since the number of samples is $\Theta(S \log(1/\rho))$, we get that it holds for some $j$ with probability $1 - \rho$. $\square$

**Lemma D.2.** *Let $E^s \in \mathbb{R}_+^S$ be the vector of the minimum expected times to reach $s$ from $s' \in \mathcal{S}$ in true MDP $\mathcal{M}$, i.e., $E_{s'}^s = \min_\pi T_{s' \to s}^\pi$. Note that $E_s^s = 0$. For any episode $k$, if for every $s, a$ there exists some $j$ such that*

$$
Q_{s,a}^{j,k} \cdot E^s \leq P_{s,a} \cdot E^s + \delta, \tag{15}
$$

*for some $\delta \in [0, 1)$, then the diameter of extended MDP $\tilde{\mathcal{M}}^k$ is at most $\frac{D}{1-\delta}$, where $D$ is the diameter of MDP $\mathcal{M}$.*

*Proof.* Fix a $k$. For brevity, we omit the superscript $k$ in below.

Fix any two states $s_1 \neq s_2$. We prove the lemma statement by constructing a policy $\tilde{\pi}$ for $\tilde{\mathcal{M}}$ such that the expected time to reach $s_2$ from $s_1$ is at most $\frac{D}{1-\delta}$. Let $\pi$ be the policy for MDP $\mathcal{M}$ for which the expected time to reach $s_2$ from $s_1$ is at most $D$ (since $\mathcal{M}$ has diameter $D$, such a policy exists). Let $E$ be the $|S| - 1$ dimensional vector of expected times to reach $s_2$ from every state, except $s_2$ itself, using $\pi$ ($E$ is the sub-vector formed by removing $s_2^{th}$ coordinate of vector $E^{s_2}$ where $E^s$ was defined in the lemma statement. Note that $E_{s_2}^{s_2} = 0$). By first step analysis, $E$ is a solution of:

$$
E = \mathbf{1} + P_\pi^\dagger E,
$$

where $P_\pi^\dagger$ is defined as the $(S-1) \times (S-1)$ transition matrix for policy $\pi$, with the $(s, s')^{th}$ entry being the transition probability $P_{s, \pi(s)}(s')$ for all $s, s' \neq s_2$. Also, by choice of $\pi$, $E$ satisfies

$$E_{s_1} \leq D.$$

Now, we define $\tilde{\pi}$ using $\pi$ as follows: For any state $s \neq s_2$, let $a = \pi(s)$ and $j^{th}$ sample satisfies the property (15) for $s, a, E^{s_2}$, then we define $\tilde{\pi}(s) := a^j$. Let $Q_{\tilde{\pi}}$ be the transition matrix (dimension $S \times S$) for this policy.

$Q_{\tilde{\pi}}$ defines a Markov chain. Next, we modify this Markov chain to construct an absorbing Markov chain with a single absorbing state $s_2$. Let $Q_{\tilde{\pi}}^\dagger$ be the submatrix $(S-1) \times (S-1)$ submatrix of $Q_{\tilde{\pi}}$ obtained by removing the row and column corresponding to the state $s_2$. Then $Q'$ is defined as (an appropriate reordering of) the following matrix:

$$Q_{\tilde{\pi}}' = \begin{bmatrix} Q_{\tilde{\pi}}^\dagger & \mathbf{q} \\ \mathbf{0} & 1 \end{bmatrix}$$

where $\mathbf{q}$ is an $(S-1)$-length vector such that the rows of $Q_{\tilde{\pi}}'$ sum to 1. Since the probabilities in $Q_{\tilde{\pi}}$ were drawn from Dirichlet distribution, they are all strictly greater than 0 and less than 1. Therefore each row-sum of $Q_{\tilde{\pi}}^\dagger$ is strictly less than 1, so that the vector $\mathbf{q}$ has no zero entries and the Markov chain is indeed an absorbing chain with single absorbing state $s_2$. Then we notice that $(I - Q_{\tilde{\pi}}^\dagger)^{-1}$ is precisely the fundamental matrix of this absorbing Markov chain and hence exists and is non-negative (see Grinstead and Snell [2012], Theorem 11.4). Let $\tilde{E}$ be defined as the $S-1$ dimensional vector of expected time to reach $s_2$ from $s' \neq s_2$ in MDP $\tilde{\mathcal{M}}^k$ using $\tilde{\pi}$. Then, it is same as the expected time to reach the absorbing state $s_2$ from $s' \neq s_2$ in the Markov chain $Q_{\tilde{\pi}}'$, given by

$$\tilde{E} = (I - \bar{Q}_{\tilde{\pi}}^\dagger)^{-1} \mathbf{1}.$$

Then using (15) (since $E_{s_2}^{s_2} = 0$, the inequality holds for $P^\dagger, Q^\dagger$),

$$E = \mathbf{1} + P_\pi^\dagger E \geq \mathbf{1} + Q_{\tilde{\pi}}^\dagger E - \delta \mathbf{1} \implies (I - Q_{\tilde{\pi}}^\dagger) E \geq (1 - \delta)\mathbf{1}. \qquad (16)$$

Multiplying the non-negative matrix $(I - Q_{\tilde{\pi}}^\dagger)^{-1}$ on both sides of this inequality, it follows that

$$E \geq (1 - \delta)(I - Q_{\tilde{\pi}}^\dagger)^{-1} \mathbf{1} = (1 - \delta)\tilde{E}$$

so that $\tilde{E}_{s_1} \leq \frac{1}{(1-\delta)} E_{s_1} \leq \frac{D}{1-\delta}$, proving that the expected time to reach $s_2$ from $s_1$ using policy $\tilde{\pi}$ in MDP $\tilde{\mathcal{M}}^k$ is at most $\frac{D}{1-\delta}$.

$\square$

# E  Useful deviation inequalities

**Fact 3** (Bernstein's Inequality, from Seldin et al. [2012] Lem 11/Cor 12). *Let $Z_1, Z_2, ..., Z_n$ be a bounded martingale difference sequence so that $|Z_i| \leq K$ and $\mathbb{E}[Z_i | \mathcal{F}_{i-1}] = 0$. Define $M_n = \sum_{i=1}^n Z_i$ and $V_n = \sum_{i=1}^n \mathbb{E}[(Z_i)^2 | \mathcal{F}_{i-1}]$. For any $c > 1$ and $\delta \in (0, 1)$, with probability greater than $1 - \delta$, if*

$$\sqrt{\frac{\ln \frac{2\nu}{\delta}}{(e-2)V_n}} \leq \frac{1}{K}$$

*then*

$$|M_n| \leq (1 + c)\sqrt{(e-2)V_n \ln \frac{2\nu}{\delta}},$$

*otherwise,*

$$|M_n| \leq 2K \ln \frac{2\nu}{\delta},$$

*where*

$$\nu = \left\lceil \frac{\ln\left(\sqrt{\frac{(e-2)n}{\ln \frac{2}{\delta}}}\right)}{\ln c} \right\rceil + 1.$$

**Corollary E.1** (to Bernstein's Inequality above). *Let $Z_i$ for $i = 1, \cdots, n$, $M_n$, and $V_n$ as above. For $n \geq 96$ and $\delta \in (0, 1)$, with probability greater than $1 - \delta$,*

$$|M_n| \leq 2\sqrt{V_n \ln \frac{n}{\delta}} + 3K \ln \frac{n}{\delta}.$$

*Proof.* Applying Bernstein's Inequality above with $c = 1 + \frac{4}{n}$, with probability greater than $1 - \delta$,

$$
\begin{aligned}
|M_n| &\leq (1+c)\sqrt{(e-2)V_n \ln \frac{2\nu}{\delta}} + 2K \ln \frac{2\nu}{\delta} \\
&\leq (1+c)\sqrt{(e-2)V_n \ln \frac{n^{\frac{4}{3}}}{\delta}} + 2K \ln \frac{n^{\frac{4}{3}}}{\delta} \\
&\leq (1+c)\sqrt{(e-2)\frac{4}{3}V_n \ln \frac{n}{\delta}} + 3K \ln \frac{n}{\delta} \\
&\leq 2\sqrt{V_n \ln \frac{n}{\delta}} + 3K \ln \frac{n}{\delta}
\end{aligned}
$$

where

$$\nu = \lceil \frac{\ln \left(\sqrt{\frac{(e-2)n}{\ln \frac{2}{\delta}}}\right)}{\ln c}\rceil + 1 = \lceil \frac{n}{2} \ln \left(\sqrt{\frac{(e-2)n}{\ln \frac{2}{\delta}}}\right)\rceil + 1 \leq \frac{n}{2} \ln \left(\sqrt{\frac{(e-2)n}{\ln 2}}\right) + 2 \leq \frac{1}{2}n^{\frac{4}{3}}.$$

$\square$

**Fact 4** (Multiplicative Chernoff Bound, Kleinberg et al. [2008] Lemma 4.9). *Consider $n$ i.i.d. random variables $X_1, \cdots, X_n$ on $[0, 1]$. Let $\mu$ be their mean and let $X$ be their average. Then for any $\alpha > 0$ the following holds:*

$$P(|X - \mu| < r(\alpha, X) < 3r(\alpha, \mu)) > 1 - e^{\Omega(\alpha)},$$

*where $r(\alpha, x) = \sqrt{\frac{\alpha x}{n}} + \frac{\alpha}{n}$.*

*More explicitly, we have that with probability $1 - \rho$,*

$$|X - \mu| < \sqrt{\frac{3 \log(2/\rho)X}{n}} + \frac{3 \log(2/\rho)}{n}.$$

**Fact 5** (Cantelli's Inequality). *Let $X$ be a real-valued random variable with expectation $\mu$ and variance $\sigma^2$. Then $P(X - \mu \geq \lambda) \leq \frac{\sigma^2}{\sigma^2 + \lambda^2}$ for $\lambda > 0$ and $P(X - \mu \geq \lambda) \geq 1 - \frac{\sigma^2}{\sigma^2 + \lambda^2}$ for $\lambda < 0$.*

**Fact 6** (Berry-Esseen Theorem). *Let $X_1, X_2, ..., X_n$ be independent random variables with $\mathbb{E}[X_i] = 0$, $\mathbb{E}[X_i^2] = \sigma_i^2 > 0$, and $\mathbb{E}[|X_i|^3] = \rho_i < \infty$. Let*

$$S_n = \frac{X_1 + X_2 + ... + X_n}{\sqrt{\sigma_1^2 + ... + \sigma_n^2}}$$

*and denote $F_n$ the cumulative distribution function of $S_n$ and $\Phi$ the cumulative distribution function of the standard normal distribution. Then for all $n$, there exists an absolute constant $C_1$ such that*

$$sup_{x \in R}|F_n(x) - \Phi(x)| \leq C_1 \psi_1$$

*where $\psi_1 = (\sum_{i=1}^{n} \sigma_i^2)^{-1/2} \max_{1 \leq i \leq n} \frac{\rho_i}{\sigma_i^2}$. The best upper bound on $C_1$ known is $C_1 \leq 0.56$ (see Shevtsova [2010]).*

**Fact 7** (Abramowitz and Stegun [1964] 26.5.21). *Consider the regularized incomplete Beta function $I_z(a, b)$ (cdf) for the Beta random variable with parameters $(a, b)$. For any $z$ such that $(a + b - 1)(1 - z) \geq 0.8$, $I_z(a, b) = \Phi(y) + \epsilon$, with $|\epsilon| < 0.005$ if $a + b > 6$. Here $\Phi$ is the standard normal CDF with*

$$y = \frac{3[w_1(1 - \frac{1}{9b}) - w_2(1 - \frac{1}{9a})]}{[\frac{w_1^2}{b} + \frac{w_2^2}{a}]^{1/2}},$$

*where $w_1 = (bz)^{1/3}$ and $w_2 = [a(1 - z)]^{1/3}$.*

The following lemma uses the above fact to lower bound the probability of a Beta random variable to exceed its mean by a quantity close to its standard deviation.

**Lemma E.2** (Anti-concentration for Beta Random Variables). *Let $F_{a,b}$ denote the cdf of a Beta random variable with parameter $(a, b)$, with $a \geq 6, b \geq 6$. Let $z = \frac{a}{a+b} + C\sqrt{\frac{ab}{(a+b)^2(a+b+1)}} + \frac{C}{a+b}$, with $C \leq 0.5$. Then,*

$$1 - F_{(a,b)}(z) \geq 1 - \Phi(1) - 0.005 \geq 0.15.$$

*Proof.* Let $x = C\sqrt{\frac{ab}{(a+b+1)}} + C$. Then, $z = \frac{a+x}{a+b}, w_1 = (b(a+x)/(a+b))^{1/3}$ and $w_2 = [a(b-x)/(a+b))]^{1/3}$. Also, $z \leq 2C\sqrt{\frac{ab}{a+b}}$. Also, $(a+b-1)(1-z) \geq (a+b-1)(1 - \frac{a}{a+b} - C\sqrt{\frac{ab}{(a+b)^2(a+b+1)}} - \frac{C}{a+b}) = (a+b-1)(\frac{b}{a+b} - \frac{C}{a+b}\sqrt{\frac{ab}{a+b+1}} - \frac{C}{a+b}) \geq \frac{a+b-1}{a+b}(b - C\sqrt{\frac{ab}{a+b+1}} - \frac{C}{a+b}) \geq \frac{11}{12}(b - C\sqrt{b} - \frac{C}{12}) \geq 0.8$. Hence we can apply Fact 7 relating Beta with Normal. We bound the numerator and denominator in the expression of $y$, to show that the relation $I_z(a, b) \leq \Phi(y) + \epsilon$ holds for some $y \leq 1$.

$$
\begin{aligned}
numerator(y) &= 3[w_1(1 - \frac{1}{9b}) - w_2(1 - \frac{1}{9a})] \\
&= 3(\frac{ab}{a+b})^{\frac{1}{3}}[(1 + \frac{x}{a})^{\frac{1}{3}}(1 - \frac{1}{9b}) - (1 - \frac{x}{b})^{\frac{1}{3}}(1 - \frac{1}{9a})] \\
&\leq 3(\frac{ab}{a+b})^{\frac{1}{3}}[(1 + \frac{x}{3a})(1 - \frac{1}{9b}) - (1 - \frac{x}{3b} - \frac{2x^2}{9b^2})(1 - \frac{1}{9a})] \\
&= 3(\frac{ab}{a+b})^{\frac{1}{3}}[(\frac{b-a}{9ab}) + (\frac{x(a+b)}{3ab}) - (\frac{2x}{27ab})] + 3(\frac{ab}{a+b})^{\frac{1}{3}}[\frac{2x^2}{9b^2}(1 - \frac{1}{9a})] \\
&\leq 3(\frac{ab}{a+b})^{\frac{1}{3}}[(\frac{b-a}{9ab}) + (\frac{x(a+b)}{3ab})] + 3(\frac{ab}{a+b})^{\frac{1}{3}}[\frac{2x^2}{9b^2}(1 - \frac{1}{9a})] \\
&= (\frac{ab}{a+b})^{\frac{1}{3}}(\frac{a+b}{ab})[(\frac{b-a}{3(a+b)}) + x + \frac{2x^2}{3b^2}(1 - \frac{1}{9a})] \\
&\leq (\frac{ab}{a+b})^{\frac{1}{3}}(\frac{a+b}{ab})[(\frac{b-a}{3(a+b)}) + \frac{2x^2}{3b^2}(1 - \frac{1}{9a}) + C + C(\frac{ab}{a+b})^{\frac{1}{2}}] \\
&\leq (\frac{b-a}{3\sqrt{ab(a+b)}} + \frac{4C^2\sqrt{ab}}{b^2\sqrt{a+b}} + \frac{C\sqrt{a+b}}{\sqrt{ab}} + C)(\frac{ab}{a+b})^{\frac{5}{6}}(\frac{a+b}{ab}) \\
&\leq (\frac{1}{3\sqrt{6}} + \frac{1}{6\sqrt{6}} + \frac{1}{2\sqrt{3}} + \frac{1}{2})(\frac{ab}{a+b})^{\frac{5}{6}}(\frac{a+b}{ab}).
\end{aligned}
$$

In above, we used that $C \leq \frac{1}{2}$ and $a, b \geq 6$. Similarly,

$$
\begin{aligned}
denominator(y) &= [\frac{w_1^2}{b} + \frac{w_2^2}{a}]^{1/2} \\
&= (\frac{ab}{a+b})[\frac{(1 + \frac{x}{a})^{\frac{2}{3}}}{b} + \frac{(1 - \frac{x}{b})^{\frac{2}{3}}}{a}]^{\frac{1}{2}} \\
&\geq (\frac{ab}{a+b})^{\frac{1}{3}}[\frac{(1 + \frac{2x}{3a} - \frac{x^2}{9a^2})}{b} + \frac{(1 - \frac{2x}{3b})}{a} - \frac{x^2}{9a^2}]^{\frac{1}{2}} \\
&= (\frac{ab}{a+b})^{\frac{1}{3}}[\frac{a(1 + \frac{2x}{3a} - \frac{x^2}{9a^2}) + b(1 - \frac{2x}{3b} - \frac{x^2}{9b^2})}{ab}]^{\frac{1}{2}} \\
&= (\frac{ab}{a+b})^{\frac{1}{3}}(\frac{a+b}{ab}(1 - \frac{x^2}{9ab}))^{\frac{1}{2}} \\
&\geq (\frac{ab}{a+b})^{\frac{1}{3}}(\frac{a+b}{ab}(1 - \frac{4C^2}{9(a+b)}))^{\frac{1}{2}} \\
&\geq (\frac{ab}{a+b})^{\frac{1}{3}}(\frac{a+b}{ab}(\frac{107}{108}))^{\frac{1}{2}}.
\end{aligned}
$$

Hence we have that $y \leq \frac{\frac{1}{3\sqrt{6}}+\frac{1}{6\sqrt{6}}+\frac{1}{2\sqrt{3}}+\frac{1}{2}}{\sqrt{\frac{107}{108}}} \leq 1$, so that $I_z(a,b) \leq \phi(1) + \epsilon$ for $\epsilon \leq 0.005$. The lemma statement follows by observing that $1 - F_{(a,b)}(z) = 1 - I_z(a,b) \geq 1 - \phi(1) - \epsilon \geq 1 - 0.845 - 0.005 \geq 0.15$. $\qquad\square$

**Definition 5.** *For any $X$ and $Y$ real-valued random variables, $X$ is stochastically optimistic for $Y$ if for any $u : R \to R$ convex and increasing $\mathbb{E}[u(X)] \geq \mathbb{E}[u(Y)]$.*

**Lemma E.3** (Gaussian vs Dirichlet optimism, from Osband et al. [2014] Lemma 1). *Let $Y = P^T V$ for $V \in [0,1]^S$ fixed and $P \sim Dirichlet(\alpha)$ with $\alpha \in R_+^S$ and $\sum_{i=1}^S \alpha_i \geq 2$. Let $X \sim N(\mu, \sigma^2)$ with $\mu = \frac{\sum_{i=1}^S \alpha_i V_i}{\sum_{i=1}^S \alpha_i}$, $\sigma^2 = (\sum_{i=1}^S \alpha_i)^{-1}$, then $X$ is stochastically optimistic for $Y$.*

**Lemma E.4** (Gaussian vs Beta optimism, Osband et al. [2014] Lemma 6). *Let $\tilde{Y} \sim Beta(\alpha, \beta)$ for any $\alpha, \beta > 0$ and $X \sim N(\frac{\alpha}{\alpha+\beta}, \frac{1}{\alpha+\beta})$. Then $X$ is stochastically optimistic for $\tilde{Y}$ whenever $\alpha + \beta \geq 2$.*

**Lemma E.5** (Dirichlet vs Beta optimism, Osband et al. [2014] Lemma 5). *Let $y = p^T v$ for some random variable $p \sim Dirichlet(\alpha)$ and constants $v \in \mathcal{R}^d$ and $\alpha \in \mathcal{N}^d$. Without loss of generality, assume $v_1 \leq v_2 \leq \cdots \leq v_d$. Let $\tilde{\alpha} = \sum_{i=1}^d \alpha_i(v_i - v_1)/(v_d - v_1)$ and $\tilde{\beta} = \sum_{i=1}^d \alpha_i(v_d - v_i)/(v_d - v_1)$. Then, there exists a random variable $\tilde{p} \sim Beta(\tilde{\alpha}, \tilde{\beta})$ such that, for $\tilde{y} = \tilde{p}v_d + (1 - \tilde{p})v_1$, $\mathbb{E}[\tilde{y}|y] = \mathbb{E}[y]$.*

**Lemma E.6.** *If $\mathbb{E}[X] = \mathbb{E}[Y]$ and $X$ is stochastically optimistic for $Y$, then $-X$ is stochastically optimistic for $-Y$.*

*Proof.* By Lemma 3.3 in Osband et al. [2014], $X$ stochastically optimistic for $Y$ is equivalent to having $X =_D Y + A + W$ with $A \geq 0$ and $\mathbb{E}[W|Y + A] = 0$ for all values $y + a$. Taking expectation of both sides, we get that $\mathbb{E}[X] = \mathbb{E}[Y] + \mathbb{E}[A] + \mathbb{E}[W]$ and since $\mathbb{E}[X] = \mathbb{E}[Y] = 0$ and $\mathbb{E}[W] = \mathbb{E}[\mathbb{E}[W|Y + A]] = 0$ we get that $\mathbb{E}[A] = 0$. Since $A \geq 0$, $A = 0$. Also note that $\mathbb{E}[W|Y = y] = 0$ for all $y$.

Now we can show that $-X$ is stochastically optimistic for $-Y$ as follows: From above, $-X =_D -(Y + A + W) = -Y + (-W)$. Then for all $y'$, $\mathbb{E}[-W| - Y = y'] = -\mathbb{E}[W|Y = -y'] = 0$ by definition of $W$. Therefore, $-X$ is stochastically optimistic for $-Y$. $\qquad\square$

**Corollary E.7.** *Let $Y$ be any distribution with mean $\mu$ such that $X \sim N(\mu, \sigma^2)$ is stochastically optimistic for $Y$. Then with probability $1 - \rho$,*

$$|Y - \mu| \leq \sqrt{2\sigma^2 \log(2/\rho)}.$$

*Proof.* For any $s > 0$, and $t$, and applying Markov's inequality,

$$P(Y - \mu > t) = P(Y > \mu + t) = P(e^{sY} > e^{s(\mu+t)}) \leq \frac{\mathbb{E}[e^{sY}]}{e^{s(\mu+t)}}.$$

By Definition 5, taking $u(a) = e^{sa}$, which is a convex and increasing function, $\mathbb{E}[e^{sY}] \leq \mathbb{E}[e^{sX}]$, and hence

$$P(Y - \mu > t) \leq \frac{\mathbb{E}[e^{sX}]}{e^{s(\mu+t)}} = \frac{e^{\mu s + \frac{1}{2}\sigma^2 s^2}}{e^{s(\mu+t)}} = e^{\frac{1}{2}\sigma^2 s^2 - st}.$$

Since the above holds for all $s > 0$, using $s = \frac{t}{\sigma^2}$, $P(Y - \mu > t) \leq e^{-\frac{t^2}{2\sigma^2}}$.

Similarly, for the lower tail bound, we have for any $s > 0$,

$$P(Y - \mu < -t) = P(-Y > -\mu + t) = P(e^{s(-Y)} > e^{s(-\mu+t)}) \leq \frac{\mathbb{E}[e^{s(-Y)}]}{e^{s(-\mu+t)}}.$$

By Lemma E.6, $-X$ is stochastically optimistic for $-Y$, so $\mathbb{E}[e^{s(-Y)}] \leq \mathbb{E}[e^{s(-X)}]$, and hence

$$P(Y - \mu < -t) \leq \frac{\mathbb{E}[e^{s(-X)}]}{e^{s(-\mu+t)}} = \frac{e^{-\mu s + \frac{1}{2}\sigma^2 s^2}}{e^{s(-\mu+t)}} = e^{\frac{1}{2}\sigma^2 s^2 - st}.$$

Again letting $s = \frac{t}{\sigma^2}$, $P(Y - \mu < -t) \leq e^{-\frac{t^2}{2\sigma^2}}$.

Then, for $t = \sqrt{2\sigma^2 \log(2/\rho)}$, we have that

$$P(|Y - \mu| \le \sqrt{2\sigma^2 \log(2/\rho)}) \ge 1 - \rho.$$

$\square$

**Lemma E.8** (Binomial, Multinomial). *Let $\hat{Y} = \hat{p}^T v$ where $\hat{p} \in \Delta^S$ be distributed as multinomial average with parameter $n, p$ and fixed $v \in R^d$, where $0 \le v_i \le D$. Then, there exists a random variable distributed as $\hat{q} \sim \frac{1}{n} Binomial(n, \frac{p^T h}{D})$ such that, $\mathbb{E}[\hat{q}|\hat{Y}] = \frac{1}{D}\hat{Y}$.*

*Proof.* Let $X_i^j, j = 1, \ldots, n$ denote the outcomes of the trials used to define $\hat{p}_i$, that is,

$$\hat{p}_i := \sum_{j=1}^{n} X_i^j / n$$

where $X_i^j, j = 1, \ldots, n$ are distributed as $X_i^j \sim Multivariate(p, 1)$.

For every $i$, define $n$ i.i.d. variables $Y_i^j, j = 1, \ldots, n$, where $Y_i^j \sim Bernoulli(v_i/D)$, and is independent of $X_i^j$. Define $\hat{q}$ as:

$$\hat{q} = \frac{1}{n} \sum_i \sum_{j=1}^{n} X_i^j Y_i^j / n$$

Let $\mathcal{X} = \{X_{i,j}, i = 1, \ldots, S, j = 1, \ldots, n\}$. Then,

$$
\begin{aligned}
\mathbb{E}[\hat{q}|\hat{p}^T v, n] &= \mathbb{E}[\mathbb{E}[\hat{q}|\mathcal{X}, \hat{p}^T v, n]|\hat{p}^T v, n] \\
&= \mathbb{E}[\mathbb{E}[\hat{q}|\mathcal{X}, n]|\hat{p}^T v, n] \\
&= \frac{1}{n}\mathbb{E}[\mathbb{E}[\sum_{i,j} X_i^j Y_i^j |\mathcal{X}, n]|\hat{p}^T v, n] \\
&= \frac{1}{n}\mathbb{E}[\sum_{i,j} X_i^j \mathbb{E}[Y_i^j]|\hat{p}^T v, n] \\
&= \frac{1}{n}\mathbb{E}[\sum_{i,j} X_i^j \frac{v_i}{D}|\hat{p}^T v, n] \\
&= \hat{p}^T v/D.
\end{aligned}
$$

Also, $n\hat{q}$ is a binomial random variable $Binomial(n, \frac{1}{D}p^T v)$ since it is formed by sum of outcomes of $n$ trials $\sum_{j=1}^{n} Z^j$, where each trail $Z^j = \sum_i X_i^j Y_i^j$ is an independent Bernoulli trial: takes value 1 with probability $\sum_i p_i v_i/D$. $\square$

**Corollary E.9.** *For $X = D\hat{q}$, $Y = \hat{p}^T v$ (with $\hat{q}$ and $\hat{p}^T v$ as defined in the previous lemma), $X$ is stochastically optimistic for $Y$.*

*Proof.* We have

$$\mathbb{E}[X - Y|Y] = \mathbb{E}[D\hat{q} - \hat{p}^T v|\hat{p}^T v] = 0.$$

Then stochastic optimism follows from applying the optimism equivalence condition from Lemma 3 (Condition 3) of Osband et al. [2014]. $\square$