[Reviews · NeurIPS 2017]

Reviewer 1



Posterior Sampling for Reinforcement Learning: Worst-Case Regret Bounds ======================================================================= This paper presents a new algorithm for efficient exploration in Markov decision processes. This algorithm is an optimistic variant of posterior sampling, similar in flavour to BOSS. The authors prove new performance bounds for this approach in a minimax setting that are state of the art in this setting. There are a lot of things to like about this paper: - The paper is well written and clear overall. - The analysis and rigour is of a high standard. I would say that most of the key insights do come from the earlier "Gaussian-Dirichlet dominance" of Osband et al, but there are some significant extensions and results that may be of wider interest to the community. - The resulting bound, which is O(D \sqrt{SAT}) for large enough T is the first to match the lower bounds in this specific setting (communicating MDPs with worst-case guarantees) up to logarthmic factors. Overall I think that this is a good paper and will be of significant interest to the community. I would recommend acceptance, however there are several serious issues that need to be addressed before a finalized version: - This paper doesn't do a great job addressing the previous Bayesian analyses of PSRL, and the significant similarities in the work. Yes, these new bounds are "worst case" rather than in expectation and for communicating MDPs instead of finite horizon... however these differences are not so drastic to just be dismissed as incomparable. For example, this paper never actually mentions that PSRL/RLSVI used this identical stochastic optimism condition to perform a similar reduction of S to \sqrt{S}. - Further, this algorithm is definitively not "PSRL" (it's really much more similar to BOSS) and it needs a specific name. My belief is that the current analysis would not apply to PSRL, where only one sample is taken, but PSRL actually seems to perform pretty well in experiments (see Osband and Van Roy 2016)... do the authors think that these extra samples are really necessary? - The new bounds aren't really an overall improvement in big O scaling apart from very large T. I think this blurs a lot of the superiority of the "worst-case" analysis over "expected" regret... none of these "bounds" are exactly practical, but their main value is in terms of the insight towards efficient exploration algorithms. It seems a significant downside to lose the natural simplicity/scalability of PSRL all in the name of these bounds, but at a huge computational cost and an algorithm that no longer really scales naturally to domains with generalization. - It would be nice to add some experimental evaluation of this new algorithm, perhaps through the code published as part of (Osband and Van Roy, 2016)... especially if this can add some intuition/insight to the relative practical performance of these algorithms. ============================================================================== = Rebuttal comments ============================================================================== Thanks for the author feedback, my overall opinion is unchanged and I definitely recommend acceptance... but I *strongly* hope that this algorithm gets a clear name separate to PSRL because I think it will only lead to confusion in the field otherwise. On the subject of bounds I totally agree these new bounds (and techniques) are significant. I do believe it would give some nice context to mention that at a high level the BayesRegret analysis for PSRL also reduced the S -> sqrt{S}. On the subject of multiple samples the authors mention “An Optimistic Posterior Sampling Strategy for Bayesian Reinforcement Learning, Raphael Fonteneau, Nathan Korda, Remi Munos”, which I think is a good reference. One thing to note is that this workshop paper (incorrectly) compared a version of PSRL (and optimistic PSRL) that resamples every timestep... this removes the element of "deep exploration" due to sampling noise every timestep. If you actually do the correct style of PSRL (resampling once per episode) then multiple samples does not clearly lead to improved performance (see Appendix D.3 https://arxiv.org/pdf/1607.00215.pdf ) On the subject of bounds and large T, I do agree with the authors, but I think they should make sure to stress that for small T these bounds are not as good and whether these "trailing" terms are real or just an artefact of loose analysis.

Reviewer 2



The authors present a reinforcement learning algorithm based on posterior sampling and show that it achieves a worst-case total regret bound that is an improvement over the best available bound by a factor of sqrt(S) for the case of infinite horizon average rewards, thus closing the gap between the known lower bound and achievable upper bound. The paper is easy to read and technically sound. Although I did not go through the detailed proofs, the argument of how the regret bound is improved via Dirichlet concentration is compelling. The author should provide some details on the differences between the techniques used in proving the worst-case regret bounds compared to those used in proving the existing Bayesian-regret bounds. Some empirical comparisons between the proposed algorithm and UCRL2 would be helpful in demonstrating the performance difference afforded via posterior sampling.